



# The development and evaluation of a hydrological seasonal forecast system prototype for predicting spring flood volumes in Swedish rivers

Kean Foster[1, 2], Cintia Bertacchi Uvo[2], Jonas Olsson[1]

[1]Research & Development (hydrology), Swedish Meteorological and Hydrological Institute, 601 76 Norrköping, Sweden
[2]Department of Water Resources Engineering, Lund University, Box 118, 221 00 Lund, Sweden

*Correspondence to*: Kean Foster (kean.foster@smhi.se)

**Abstract.** Hydropower makes up nearly half of Sweden's electrical energy production. However, the distribution of the water resources is not aligned with demand, most of the inflows to the reservoirs occur during the spring flood period. This means that carefully planned reservoir management is required to help redistribute the water resources to ensure optimal production and accurate forecasts of the spring flood volume (SFV) is essential for this. The current operational SFV forecasts use a historical ensemble approach where the HBV model is forced with historical observations of precipitation and temperature. In this work we develop and test a multi-model prototype, building on previous work, and evaluate its ability to forecast the SFV in 84 sub-basins in northern Sweden. The testing is done using cross-validated hindcasts for the period 1981-2015 and the results are evaluated against both climatology and the current system to determine skill. Both the considered multi-model methods considered showed skill over the reference forecasts, however the version that combined the historical modelling chain, dynamical modelling chain, and statistical modelling chain performed better than the other and was chosen for the prototype. The prototype was able to outperform the current operational system on average 65% of the time and reduce the error in the SFV by ~6% across all subbasins and forecast dates.

## 1 Introduction

The spring flood period (sometimes referred to as the spring melt or freshet period in the literature) is of great importance in snow dominated regions like Sweden where hydropower accounts for nearly half of the country's electrical energy production (SCB, 2016). Between 55-70% of the annual inflows to reservoirs in the larger hydropower producing rivers occur during this relatively short period, typically from mid-April/early-May to the end of July. This means that the majority of the annual water resources available for hydropower production would only be available to producers during this period if it were not regulated through carefully planned reservoir management. This reservoir management is important as the energy demand is out of phase with the natural availability of the water resources; typically demand is higher (lower) during the colder (warmer) months when the inflows are lower (higher). Therefore the goal is to redistribute the availability of these resources from the spring flood period to other times of the year when electricity demand is higher i.e. during the colder





winter half year, while maintaining a balance between a sufficiently large volume of water for optimal production and enough remaining capacity for safe flood risk management (Olsson et al., 2016). The typical strategy for operators in Sweden is to have reservoirs at around 90% capacity at the end of the spring flood which is then ideally maintained until the beginning of winter. To achieve this operators require reliable seasonal forecast information to help them in planning the

operations both leading up to and during the spring flood period.

The sources of predictability for hydrological seasonal forecasts come from the initial hydrological conditions i.e. information relating to the water stores within in the catchment (e.g. Wood and Lettenmaier, 2008; Wood et al. 2015; Yossef et al., 2013), and also from knowledge of the weather during the forecast period i.e. seasonal meteorological forecasts (e.g. Bennet et al., 2016; Doublas-Reyes et al., 2103; Wood et al. 2015; Yossef et al., 2013). Hydrological seasonal forecasts

attempt to leverage at least one of these sources of predictability to make skilful predictions of future streamflow.

In practice there are two predominant approaches to making hydrological forecasts at the seasonal scale; statistical approaches and dynamical approaches. Statistical approaches utilise empirical relationships between predictors and a predictand, typically streamflow or a derivative thereof (e.g. Garen, 1992; Pagano et al., 2009). These predictors can vary greatly in type from local hydrological storage variables like snow and groundwater storages (e.g. Robertson et al. 2013;

Rosenberg et al., 2011), to local and regional meteorological variables (e.g. Còrdoba-Machado et al., 2016; Olsson et al., 2016), to large scale climate data such as ENSO indices (e.g. Schepen et al., 2016; Shamir, 2017). All, however, are trying to leverage the predictability in these predictors that originate from one of the two aforementioned sources. Dynamical approaches use a hydrological model, typically initialised with observed data up to the forecast date so that the model state is a reasonable approximation of the initial hydrological conditions, and then force it with either historical observations or force

it using data representative of the future meteorological conditions such as general circulation model (GCM) outputs (e.g. Crochemore et al. 2016; Olsson et al. 2016; Yuan et al. 2013, 2015, 2016). Attempts to improve these types of approaches have involved bias adjusting the GCM outputs (e.g. Crochemore et al., 2016, Lucatero et al., 2017; Wood et al., 2002; Yuan et al. 2015) or bias adjusting the hydrological model outputs (e.g. Lucatero et al., 2017) or a combination of both (e.g. Yuan and Wood, 2012). Another dynamical approach is the well-established ensemble streamflow prediction (ESP; e.g. Day,

1985). This is similar to the previous approach, however instead of using GCM outputs to force the hydrological model it uses an ensemble of historical data. This approach is perhaps one of the most widely used methods and is still the subject of new research. Recent work have looked at conditioning the ensembles before using them, this conditioning can be done using GCM outputs (e.g. Crochemore et al., 2016), climate indices, and circulation pattern analysis (e.g. Beckers et al., 2016; Olsson et al. 2016; Yossef et al., 2016).

The current practice at the Swedish Meteorological and Hydrological Institute (SMHI) for seasonal forecasts of reservoir inflows is the ESP approach. It assumes that historical observations of precipitation and temperature are possible representations of future meteorological conditions and are used to force the HBV hydrological model (e.g. Bergström, 1976; Lindström et al., 1997) to give an ensemble forecast that has a climatological evolution from the initial conditions. A number of attempts have been made in the past to improve the performance of these spring flood forecasts with limited





success (Arheimer et al., 2010) demonstrating that these seasonal forecasts are already of a high quality. Work by Olsson et al. (2016) on improving these forecasts was able to realise reasonable improvements using a multi-model approach. By combining a statistical approach, dynamical approach and an analogue approach (conditioned ESP) they were able to show a ~4% reduction in the forecast error of the spring flood volume (SFV). The purpose of this paper is to continue on and update

the work started by Olsson et al. (2016) to develop and evaluate a hydrological seasonal forecast system prototype for forecasting the spring flood volumes in Sweden.

This paper is organised as follows. Section 2 outlines the prototype, including the individual model chains, the experimental setup, the methods and tools used, and the study area and data used in this work. Section 3 presents and discusses the cross-validated evaluation scores for the prototype, first with reference to climatology and then with reference to the current

operational system that is in use at SMHI. Section 4 concludes with the main findings and a brief outlook for future work.

## 2 Materials and Methods

### 2.1 The multi-model system and the individual modelling chains

In this section we present the modelling approaches used in this work. These are based on those explored by Olsson et al. (2016) with some modification to facilitate their use in an operational environment. First we briefly present the multi-model

prototype (Sect. 2.1.1) followed by a brief overview of the individual modelling chains used in the multi-model and why we chose them (Sect. 2.1.2 – 2.1.5). For more information regarding the individual modelling chains readers are referred to Olsson et al. (2016) and the accompanying supplement.

### 2.1.1 The multi-model ensemble (ME)

The prototypes developed in this work builds on an approach first proposed by Foster et al. (2010) and later improved upon

and first tested by Olsson et al. (2016). The aim is to adapt their methodology for use in an operational environment and then evaluate the resulting prototype against the current operational system using cross-validated hindcasts. Four different modelling chains were considered when developing the prototype (Sect. 2.1.2 – 2.1.5). The performances of different combinations of these four were tested and it was found that the two three-model combinations, analogue-dynamic-statistical and historical-dynamic-statistical, performed best on average. A combination of all four modelling chains was not

considered as the analogue model chain is a subset of the historical model chain.

Figure 1 shows the generalised schematic of the two prototypes, $ME_{ads}$ and $ME_{hds}$ (where the subscripts refer to the individual modelling chains making up each multi-model), including where the current methodologies differ significantly from those in previous works. These differences are discussed in the relevant modelling chain sections below. The prototypes are multi-model ensembles of the outputs from the three relevant individual modelling chains. These outputs are

pooled together rather than using an asymmetric weighting scheme due to the lack of data points, a total of 35 spring flood



events, from which to derive a robust weighting scheme. The simple weighting scheme used by Olsson et al. (2016) was not considered either along similar lines of reasoning.

### 2.1.2 Historical ensemble (HE)

The historical model chain, the dark blue chain third from the left in Figure 1, is an ensemble forecast made by forcing a rainfall-runoff model with historical observations of precipitation and temperature. This approach is often referred to as ESP (ensemble streamflow prediction) in the literature but we chose not to as we feel our terminology is more descriptive in the context of this work. This is the current operational seasonal forecasting practice at SMHI. The HBV model (Bergström, 1976; Lindström et al. 1997) is initialized by using observed meteorological inputs (P and T) to force the model up to the forecast date so that the model state reflects the current hydro-meteorological conditions. Then, typically all available historical daily P and T series for the period from the forecast issue date to the end of the forecasting period are used as input to HBV, generating an ensemble of forecasts that are climatological in their evolution from the initial conditions. The HE is used as the reference ensemble unless otherwise stated.

The HBV is run one river system at a time and the model outputs are later regrouped into three clusters (Section 2.6). Typically only historical data prior to the forecast date are used to force the model, however to allow for a more robust cross validation all data including for years after the forecast date were used (excluding the year in question of course). Unfortunately, the scope of this work did not allow for the recalibration of the HBV model before each cross validated hindcast. This will potentially inflate the performance of the model for the hindcasts of years that were used in the calibration of the model. This will affect the analogue and dynamical model chains too as they also incorporate the HBV model in their setup.

### 2.1.3 Analogue ensemble (AE)

The analogue model chain, the light blue chain furthest to the left in Figure 1, is a subset of the HE. The hypothesis is that it is possible to identify a reduced set of historical years (an analogue ensemble) that describes the weather in the coming forecasting period better than the full historical ensemble used in HE. In this work the circulation pattern approach used by Olsson et al. (2016) was omitted due to data availability issues making it impractical for operational applications. Additionally, the teleconnection approach was revised to take advantage of the findings by Foster et al. (2012) and Foster et al. (2016) where they identified which teleconnection patterns are related to the SFV and for which period of their persistence prior to the spring flood this connection is strongest. The teleconnection indices they identified are the Arctic oscillation (AO) and the Scandinavian pattern (SCA) and the periods of persistence for these indices are the seven and eight months leading up to the spring flood respectively.

The persistence for each teleconnection index is calculated from the beginning of the aforementioned period to one month prior to the forecast date (a limitation imposed by data availability), similarly this was done for all years in the climatological





ensemble. If the values of these indices are considered to be coordinates in Euclidean space we defined analogue years to be those years whose positions are within a distance of 0.2 from the position of the forecast year. The selection of the analogues is done at the regional scale, by cluster (Section 2.6), and these selections applied to the associated sub-catchments in turn. Similar to the HE, the analogue method makes use of both prior and later years to the hindcast year for the cross validated

hindcasts.

### 2.1.4 Dynamical modelling ensemble (DE)

The dynamical model chain,  the dark red chain furthest to the left in Figure 1, is similar to the HE; an adequately initialised HBV model is forced by an ensemble of seasonal forecasts of daily P and T from the ECMWF IFS system 4 (Sect. 2.7 Seasonal data). A change to previous work has these daily P and T data bias adjusted first using the DBS method developed

by Yang et al. (2010) before being used to force HBV. There have been some criticisms raised lately regarding the applicability of quantile mapping for bias adjusting seasonal data (e.g. Zhao et al., 2017). They point out that although quantile mapping approaches are effective at bias correction they cannot ensure reliability in forecast ensemble spread or guarantee coherence. Unfortunately, the scope of this work did not allow the testing of other bias adjustment methods but the criticism is noted and further work is planned to address these points.

These bias adjusted data are then converted into HBV inputs by mapping them from their native grid onto the HBV sub-catchments. The mapping is done by areal weighting and the resulting sub-catchment average P and T values are then adjusted to represent different altitude fractions within the catchment. These data are then used to force the HBV model from the same initial state as used in the HE procedure.

No changes are made this methodology to accommodate the cross-validated hindcasting is done with the other model chains.

### 2.1.5 Statistical modelling ensemble (SE)

The statistical model chain, the orange chain second from the left in Figure 1, is an ensemble forecast produced by downscaling forecasted or modelled large-scale variables (predictors) to the SFV for each cluster (predictand). The downscaling is done using an SVD approach (singular variable decomposition). The predictors are three large scale circulation variables (Section 2.7) and the modelled snow depths from the HBV initial conditions. The outputted ensembles

of SFV are combined using a simple arithmetic weighting system. The normalised squared covariance between the four predictors and the predictands are ranked for each forecast initialisation date and weights between 0 and 1 are applied to the different predictors according to their rank. The lowest ranked predictor is assigned a weight of 0.1 (= 1/10), the next lowest predictor is assigned a weight of 0.2 (= 2/10), and so on until all four have been assigned a weight. The reason that an asymmetric weighting scheme is used here is that there is physical support for it. Early in the season the snowpack, which is

the majority contributor to the spring flood volume, is still a fraction of what it will be and is still accumulating. Therefore, the coming meteorological conditions, which dictate snowpack evolution, are more important earlier on in the season than





they are later giving physical support for asymmetric weighting. Additionally, the relative importance of these meteorological with respect to each other predictors differs with time too.

The relative simplicity of the statistical model chain means that it was possible to retrain the model before each hindcast during the cross-validation calculations allowing for no overlap between the calibration and validation periods.

## 2.2 Defining the spring flood

In previous works the spring flood period has often been defined in terms of calendar months e.g. May-June-July (Nilsson et al., 2008; Foster and Uvo, 2010; Arheimer et al., 2011; Olsson et al., 2016; Foster et al., 2016). This definition of the spring flood period is not ideal as it does not take into account the interannual and geographical variations in the timing of the

spring flood onset. In this work we propose an improvement to this practice where we define the spring flood to be the period from the onset date to the end of July.

We define the onset as the nearest local minima in the hydrograph before the date after which the inflows are above the 90th percentile, with reference to the inflows during the first 80 days of the current year, for a period of at least 30 days (Figure 2). For forecasts made after January i.e. those made in February, March, April, and May, the missing inflow data between

the 1 January and the forecast date are filled with simulated inflow data from the HBV model using observed precipitation and temperatures as input data.

A drawback to this definition is that it is not comprehensive as the end of the spring flood is not defined according to the hydrograph but rather by date. The reasons for not defining the end of the spring flood objectively are twofold. Firstly, the forecast horizon for the ECMWF-IFS is seven months which means that forecasts initialised in January may not encompass

the entire spring flood period, and secondly, a robust and objective definition of what constitutes the end of the spring flood was difficult to realise within the scope of this work. Further work is needed to resolve this in a more satisfactory manner.

## 2.3 Experimental setup

The challenge in this work was to perform a robust evaluation on a limited dataset (35 spring floods, 1981-2015) while minimising the risk of unstable or over fitted statistics. Therefore, a leave-one-out cross validation (LOOCV) protocol was

adopted. Additionally, as it was not practical to recalibrate the HBV model before each step of the LOOCV process; the statistical model uses the same periods for training as those used to calibrate and validate the HBV model. LOOCV is a model evaluation technique that uses n-1 data points to train the models and the data point left out is used for validation. This process is repeated n times to give a validation dataset of length n. This allowed for a more robust evaluation with a limited dataset and to be able to sample more of the variability in the training period than if a traditional validation were performed.

The second point is especially advantageous for evaluating the statistical model which is especially sensitive to situations that were not found within the training period. LOOCV was applied to the individual model chains.



To assess the relative skill for different lead times, we evaluate hindcasts issued on the 1[st] of January (Jan), 1[st] of February (Feb), 1[st] of March (Mar), 1[st] of April (Apr), and 1[st] of May (May) for the spring floods 1981-2015. The evaluation of performance is done in terms of how well the SFV is forecasted.

### 2.4 Evaluation

As it has been mentioned above, we are interested in the ability of a multi-model ensemble's ability to forecast the SFV at differing lead-times i.e. forecasts initialised on the first of the month for the months of January through May. It was suggested by Cloke and Pappenberger (2008) that for a rigorous assessment of the quality of a hydrological ensemble prediction system (HEPS) it is not only important to select appropriate verification measures but also to use several different measures so that different properties of the forecast skill can be estimated resulting in a more comprehensive evaluation.

The evaluations in this paper are designed to answer the following questions:

- Can the forecasts improve on the reference forecast error?
- How often do the forecasts perform better than the reference forecast?
- Are the forecasts better at capturing the interannual variability than the reference forecast?
- Are the forecasts better at discriminating between events and non-events than the reference forecast?
- Are the forecasts sharper than the reference forecast?
- Are the forecasts more sensitive to uncertainty that the reference forecasts?

The verification measures used to answer these questions are described below and summarised in Table 1.

### Mean absolute error skill score (MAESS)

One of the most commonly published scores, even the recommended method, when evaluating HEPS is the continuous rank probability score (CRPS, Hersbach, 2000). However, since we have a limited number of data points, only 35 cross validated hindcasts per subbasin, and that the CRPS compares distributions we deemed its use unsuitable for this work. We chose to use the mean absolute error (MAE) to evaluate general forecast performance as the CRPS collapses to the MAE for deterministic forecasts (Hersbach, 2000). Therefore by assuming the ensemble mean to be the deterministic forecast a MAE

skill score (MAESS) can be expressed as

$$MAESS = 1 - \frac{MAE_f}{MAE_r} \qquad (1)$$

where $f$ and $r$ denote forecast and reference respectively, and $MAE$ is defined as

$$MAE = \frac{1}{n} \sum_{y=1}^{n} \left| \frac{SFV_o^y - SFV_f^y}{SFV_o^y} \right| \qquad (2)$$

where $y$ denotes year, $n$ denotes the total number of years, and $o$ denotes observations. The MAESS has a range between negative infinity and 1 with positive values indicating skill over the reference forecast and a value of one a perfect forecast.





**Frequency of years (FY$^+$)**

In their work Olsson et al. (2016) proposed FY$^+$ as a complimentary performance measure to scores such as the MAESS. They are complimentary in that the MAESS is a measure of how much better the forecast is than the reference forecast while

5  FY$^+$ is the frequency or how often the forecast is better i.e. how often the absolute error is lower. FY$^+$ scores range from 0 to 100% where values above 50% indicate that the multi-model forecast has skill over the reference forecast. By assuming the ensemble mean to be the deterministic forecast FY$^+$ is expressed as

$$FY^+ = \frac{100}{n} \sum_{y=1}^{n} H^y$$

where $H$ is the Heaviside function defined by

$$H^y = \begin{cases} 0, & \mathrm{AbsE_r^y} < \mathrm{AbsE_f^y} \\ 1, & \mathrm{AbsE_r^y} > \mathrm{AbsE_f^y} \end{cases}$$

where $AE$ is the absolute error.

**Nash–Sutcliffe model efficiency (NSE)**

The NSE (Nash and Sutcliffe 1970) is a normalized statistic that determines the relative magnitude of the residual variance compared to the measured data variance. The NSE has a range from -∞ to 1 with 1 being a perfect match and values above 0 denoting that the forecast has skill over climatology. For this work it can be interpreted as how well the forecasted SFV

15  matches the observed SFV year on year and as such is complimentary to MAESS and FY$^+$. By assuming the ensemble mean to be the deterministic forecast the NSE can be expressed as

$$NSE = 1 - \frac{\sum_{y=1}^{n}\left(SFV_o^y - SFV_f^y\right)^2}{\sum_{y=1}^{n}\left(SFV_o^y - \overline{SFV_o}\right)^2} \tag{5}$$

To assess the skill of the multi-model ensemble, with respect to the reference historical ensemble, the difference in their NSE is calculated

$$\Delta NSE = NSE_f - NSE_r \tag{6}$$

where $\Delta$NSE > 0 indicates that the multi-model forecast has skill over the reference forecast.

**Relative operating characteristic skill score (ROCSS)**

The ROCSS is a skill score based on the area under the curve (AUC) in a relative operating characteristic diagram. ROCSS values below 0 indicate the forecast has no skill over climatology while values over 0 indicate skill with 1 being a perfect forecast. The ROC diagram measures the ability of the forecast ensemble to discriminate between an event and a non-event

25  given a specific threshold. For this work the ROCSS were calculated for the upper tercile (x ≥ 66.7%), middle tercile (66.7% < x ≤ 33.3%) and lower tercile (x < 33.3%). These scores estimate the skill of ensemble forecasts to distinguish between





below normal (BN), near normal (NN) and above normal (AN) anomalies. Hamill and Juras (2006) define the ROC skill score to be

$$ROCSS = 2 * AUC - 1 \qquad (7)$$

where AUC is the area under the curve when mapping hit rates against false alarm rates

$$AUC = \sum_{y=1}^{n+1} \frac{(FAR_y - FAR_{y-1})(HR_y + HR_{y-1})}{2} \qquad (8)$$

where FAR = false alarm rate and HR = hit rate. False alarms are defined as both the false positive and false negative
forecasts, or type I and type II errors. Hits are defined as correctly forecasted events.

**Inter quartile range skill score (IQRSS) and uncertainty sensitivity skill score (USS)**

Sharpness is an intrinsic attribute to HEPS, giving an indication of how large the ensemble spread is. Forecasts ensembles that are too spread are overly cautious and have limited value for an end user due to the uncertainty of the true magnitude of
the SFV, conversely ensembles that are not spread enough are overly confident and may not be a true representation of the uncertainty thus giving the end user false confidence in the forecast (Gneiting et al., 2007). For this work the sharpness is computed as the difference between the $75^{th}$ and $25^{th}$ percentiles of the forecast distribution or the inter quartile range (IQR). The IQRSS is skill score based on the IQR and is a measure of how much better i.e. sharper the forecast ensemble is over the reference ensemble, values above 0 indicate that the forecast ensemble is an improvement over the reference ensemble. The
IQRSS is expressed as

$$IQRSS = 1 - \frac{IQR_f}{IQR_r} \qquad (9)$$

As mentioned above, sharpness can be misleading. A well-designed and calibrated ensemble should give the user an idea of the uncertainty of the forecast conveyed through the relative sharpness of the ensemble. Thus it follows that the IQR should be positively correlated to the absolute forecast error; a larger (smaller) IQR would indicate to the user that there is a larger (smaller) uncertainty in the SFV forecast. The uncertainty sensitivity skill score (USS) can be expressed as the skill score of
the Spearman rank correlations between the IQR and the absolute deterministic error

$$USS = \frac{(\rho_r - \rho_f)}{(1 - \rho_r)} \qquad (10)$$

where ρ is the Spearman rank correlation.

**2.5 Uncertainty estimation**

Due to the limited sample size of data available in this work a bootstrap approach is employed to estimate the verification measures and determine whether they are statistically significant. Again due to data limitations a more circumspect
significance level is prudent due to the course nature of the resulting statistics, we chose to set the significance level at 0.1





resulting in a 90% confidence interval between the 5$^{th}$ and 95$^{th}$ percentiles. The cross-validated hindcast ensembles were sampled, allowing for repetition, 10000 times to calculate the verification measures. We define a result to be statistically significant if the 5$^{th}$ (95$^{th}$) percentile of the bootstrapped ensemble being evaluated does not overlap the 95$^{th}$ (5$^{th}$) percentile of the bootstrapped reference ensemble.

**2.6 Study area and local data**

The subbasins sub-basins used in this work are divided into three groups using the clusters defined by Foster et al. 2012 and Foster et al. 2016, namely clusters, S$^1$, S$^2$, and S$^3$. Sweden was divided into five regions of homogeneous streamflow variability; three clusters located in the northern parts of the country, where snow dominates the hydrological processes (northern group), and two located in the southern part, where rain dominates the hydrological processes (southern group).

For the purposes of this work we are interested only in the northern group. The numbers of subbasins per each of these clusters are 25, 19, and 40 respectively. The S in the cluster's designation denotes that the hydrological regimes are dominated by snow processes and the superscripts give the relative strength of the signal from these processes in the hydrological regime. During the winter months most of the precipitation that falls within these basins is stored in the form of a snowpack and does not immediately contribute to streamflow. During the warmer spring months, when the temperatures

rise above freezing, these snowpacks begin to melt, typically around mid- to late-April, which results in a period of high streamflow commonly referred to as the spring flood. We focus on forecasts of the accumulated streamflow volume during this period or spring flood volume (hereafter SFV).

For this work, 84 subbasins from seven hydropower producing rivers in northern Sweden (Figure 3) were used to for the development and testing of the multi-model prototype. These are those used in the current operational seasonal forecast

system at SMHI plus the two unregulated subbasins used by Olsson et al. (2016). Daily reservoir inflows for each subbasin are available from the SMHI archives starting from 1961 to the end of the last hydrological year; the data used in our work are for the period 1961-2015. These inflows are derived by adding the local streamflow to the change in reservoir storage then subtracting the streamflow from upstream basins i.e.

$$Q_{in} = \Delta S + Q_{local} - Q_{upstream} \tag{11}$$

Missing inflow data were filled by a multiple linear regression approach using simulated inflows for the subbasin and

observations from the surrounding subbasins as predictors. Of the 84 subbasins used in this work 68 had less than 1% missing data (50 of these had no missing data), four had 1-10% missing data, five had 20-30% missing data, four had 30-40% missing data, and three had 61%, 63% and 71% missing data respectively. As these subbasins are a part of the current operational forecast system they were included in the study despite some of them having a significant missing data fraction. The average NSE for the data used for filling was 0.70 (the NSE scores for the intervals above were 0.67, 0.75, 0.77, 0.61,

and 0.73 respectively) which suggests that this approach is acceptable.

Daily observations of precipitation and temperature data used in this work were obtained from the PTHBV dataset from SMHI (Johansson, 2002). The PTHBV dataset is a 4x4km gridded observation dataset of daily precipitation and temperature





data that has been created by optimal interpolation with elevation and wind taken into account. These data are available from 1961 to the present.

Table 2 gives a summary of some basic basin characteristics and statistics regarding the SFV as well as selected performance measures for the HBV rainfall-runoff model for the subbasins in each cluster. Although the ranges in subbasin areas in the different clusters are similar, except for the maximum in $S^3$, the SFV statistics increase with each cluster when looking from cluster $S^1$ to $S^3$. This is due to the effects that elevation and latitude have on how much snow processes dominate the hydrological regimes in each cluster. Subbasins in cluster $S^1$ are typically at a latitude and or elevation than those in clusters $S^2$ and $S^3$, similarly the subbasins in $S^2$ with respect to those in $S^3$. The ranges in the NSE and the relative MAE imply that in general the HBV model is adequately or well calibrated for most subbasins, however there are some subbasins for which the HBV model appears to not be well calibrated and for which there is some scope for improvement. This can be somewhat misleading as these data are a function of three different observations and as such can be subject to noise and uncertainties.

**2.7 Driving Data**

**Teleconnection indices**

This work uses monthly indices of the Arctic Oscillation and Scandinavian Pattern collected from the Climate Prediction Center (Climate Prediction Center, 2012) for the period October 1960 to May 2015.

**Seasonal data**

The ECMWF seasonal forecast system model from system 4 (Molteni et al., 2011) is the cycle36r4 version of ECMWF IFS (Integrated Forecast System) coupled with a 1° version of the NEMO ocean model. The seasonal forecasts from the ECMWF IFS were used in the following two different forms, a field of seasonal monthly averages as input to the statistical model and individual grid points of daily data for input into the HBV model.

The seasonal forecast averages are the seasonal means for each ensemble member of the different predictors which had a domain covering 75°W to 75°E and 80°N to 30°N (Figure 2) with a 1°x1° resolution. For each predictor only the first 15 ensemble members were used in this work. This is because the number of ensemble members is limited to 15 for the hindcast period while the operational seasonal forecast ensemble has 51 members. The predictors considered in this part of the work were the following: 850 hPa geopotential, 850 hPa temperature, 850 hPa zonal wind component, 850 hPa meridional wind component, 850 hPa specific humidity, surface sensible heat flux, surface latent heat flux, mean sea level pressure, 10m zonal wind component, 10m meridional wind component, 2m temperature, total precipitation.

The daily time series data are the ECMWF IFS seasonal forecasts of daily values of temperature (2mT) and the accumulated total precipitation (pr). These data have a resolution of 0.5°x0.5° and spans a period from 1981 to 2015 and had a domain covering 11° to 23°E and 55° to 70°N.




## 3 Results

The following section outlines and discusses the results from the cross-valuated hindcasts of the different approach's ability to hindcast the SFV for the period 1981-2015. First this evaluation is done for each system using climatology as a reference to assess their general skill. After that the more skilful of the two multi-model ensembles is evaluated using the HE as a

reference to assess any improved skill and thus added value of the multi-model ensemble approach over the current HE approach. The analysis is carried out on the cross-validated hindcasts of the SFV initialised on the 1$^{st}$ of January, February, March, April, and May.

### 3.1 Evaluating the different forecast systems against climatology

The different forecast approach's general skill to predict the SFV was estimated using MAESS, their skill to reproduce the

interannual variability was estimated using NSE, and finally the skill to discriminate between below BN, NN, and AN SFVs is estimated using ROCSS. Table 3 gives an overview of these scores across all subbasins and clusters for each initialisation month as well as the percentage of subbasins where the hindcasts outperformed climatology, the values in brackets are the percentage of subbasins where the hindcasts outperformed climatology and the result is statistically significant.

The performance measures for each of the three approaches are positively related to the relative timing of the hindcasts i.e.

hindcasts initialised in any month are generally more (less) skilful than the hindcasts initialised in the preceding (following) months. This can be expected as the further away in time from the spring flood that the hindcast is initialised, increasing lead time, the less the initial hydrological conditions contribute to predictability and the more uncertain the forcing data become (e.g. Wood et al., 2016; Arnal et al., 2017).

With respect to general skill and the ability to capture the interannual variation shown by the observations, the prototypes

tend to perform the better than HE with ME$_{hds}$ typically having the best performance. This is especially so when we consider the percentage of the subbasins where this improved performance is statistically significant. The gap between HE and the two prototypes in MAESS, NSE, and percentage of subbasins with improved performance over climatology tends to get smaller as the season progresses while the gap in the percentage of subbasins where improved performance is statistically significant appears to grow, at least early in the season.

However, if we turn our attention to the forecast's ability to discriminate between BN, NN, and AN SFVs then the HE holds an advantage over the two prototypes especially when it comes to identifying NN events from all forecast initialisation dates and, to a lesser extent, BN events for the later forecasts. The proposed prototypes are better at identifying AN events for all forecasts except those initialised in May where the ability of the HE is comparable. The advantage displayed by the HE to identify NN events is to be expected due to its climatological nature while the advantage with respect to BN events can

probably be attributed to a cold bias in the historical forcing data caused by climate change. The drop in relative skill by the prototypes in the later forecasts is in part due to their sharpness being worse than the HE in the later forecast (Section 3.3).





From these results we are now able to make an informed choice as to which prototype to proceed with, $ME_{hds}$ (hereafter referred to as the prototype unless stated otherwise). If we take all the results and rank the performances of the three methods then the prototype would rank first followed closely by $ME_{ads}$ and HE would rank third. However, all three forecast methods have been shown to be skilful at forecasting the SFV albeit a naïve skill.

### 3.2 Evaluating the prototype against HE

The frequency at which the prototype outperforms HE is estimated using $FY^+$, its general skill to predict the SFV is estimated using MAESS, its skill to reproduce the interannual variability is estimated using $\Delta NSE$, and finally its skill to discriminate between BN, NN, and AN SFVs is estimated using $\Delta ROCSS$. Figure 4 shows the bootstrapped scores for MAESS, FY+, and $\Delta NSE$ calculated for each hindcast initialisation month for the subbasins in cluster $S^3$. The medians of these bootstrapped scores are presented as a histogram; summary statistics are documented above the histogram. On the left-hand side are the max, mean, and min scores for the cluster i.e. the subbasins with the highest and lowest scores and the mean for the basin. On the right-hand side are the percentage of subbasins where the prototype outperformed HE, shows skill over HE ($n_{abs}^+$), the percentage of subbasins where the prototype shows statistically significant skill over HE ($n_{0.1}^+$), and the percentage of subbasins where HE statistically significant skill over the prototype ($n_{0.1}^-$).

**$FY^+$**

The prototype has a $FY^+ > 50\%$ for the majority of the subbasins in cluster $S^3$, ranging from 98% of the subbasins with an mean $FY^+$ of 61% for hindcasts initialised in January down to 73% of the subbasins and mean $FY^+$ of 56% for hindcasts initialised in May. These figures are similar, even a little higher, for subbasins in cluster $S^2$ while somewhat lower for cluster $S^1$. The number of subbasins for which the prototype has a statistically significant $FY^+ > 50\%$ ranges between 10% and 28% in cluster $S^3$ ($S^2 = 5$-37%, and $S^1 = 12$-16%). While the prototype has a statistically significant $FY^+ < 50\%$ (performs worse than HE more often than not) for 5% of subbasins for hindcasts initialised in April in cluster $S^3$ and 4% of subbasins in hindcasts initialised in May for cluster $S^1$.

**MAESS**

The prototype shows skill at improving the volume error hindcasted by HE for the majority of the subbasins, ranging between 65% and 100% of the subbasins in cluster $S^3$ ($S^2 = 74$-95%, and $S^1 = 64$-80%). This improvement tends to be largest for hindcasts initialised in January, mean MAESS of 0.12 ($S^2 = 0.11$, and $S^1 = 0.04$), and lowest for those in May, 0.04 ($S^2 = 0.05$, and $S^1 = 0.02$). The percentage of subbasins for which MAESS > 0 is statistically significant ranges between 10-53% for all clusters and hindcast initialisations, while the percentage of subbasins for which MAESS < 0 is statistically significant are 8% and 4% for hindcasts initialised in March and May in cluster $S^1$ and 3% for hindcasts initialised in both April and May in cluster $S^3$. These results also show that the prototype generally has a smaller MAE than HE especially for earlier hindcast initialisations and again for clusters $S^3$ and $S^2$.

**$\Delta NSE$**





The prototype shows skill at improving the representation of the interannual variability of the observed SFV again for most of the subbasins, ranging between 63% and 100% of subbasins in cluster $S^3$ ($S^2$ = 74-100%, and $S^1$ = 76-92%), and the mean ΔNSE ranges between 0.06 and 0.33 for subbasins in cluster $S^3$ ($S^2$ = 0.09 and 0.32, and $S^1$ = 0.06 and 0.15). The percentage of subbasins for which ΔNSE > 0 is statistically significant ranges between 16% and 63% for all clusters and hindcast

initialisations, while the percentage of subbasins for which ΔNSE < 0 is statistically significant are 4% for hindcasts initialised in January, March and May in cluster $S^1$, and 5% for hindcasts initialised in May in cluster $S^3$.

**ΔROCSS**

Figure 5, which has the same information presentation structure as Figure 4, shows the bootstrapped ΔROCSS for the lower
(BN), middle (NN), and upper (AN) terciles calculated for each hindcast initialisation month for the subbasins in cluster $S^3$.
The prototype shows skill over HE to discriminate between BN events and non-BN events for the majority of the subbasins in cluster $S^3$ for hindcasts initialised in January and February, 95% and 68% respectively ($S^2$ = 63%, 53% and $S^1$ = 68%, 32%) but this drops to less than half the subbasins in hindcasts initialised thereafter. The mean ΔROCSS ranges between - 0.04 and 0.14 in cluster $S^3$ ($S^2$ = -0.03 and 0.05, and $S^1$ = -0.06 and 0.02) with only statistically significant results being
found in favour of the prototype for 15% and 5% of subbasins for hindcasts initialised in January in clusters S3 and S2 respectively and in favour of HE for 4% of subbasins for hindcasts initialised in April in cluster $S^1$.
Out of the three terciles the prototype shows the least skill over HE with regards to discriminating between NN events and non-NN events. The percentage of subbasins for which the prototype outperforms HE ranges between 23% and 57% ($S^2$ = 37-63%, and $S^1$ = 24-60%) with mean ΔROCSS ranges between -0.07 and 0.01 ($S^2$ = -0.06 and 0, and $S^1$ = -0.05 and 0.02)
and no statistically significant results for any subbasins, both in favour or against the prototype.
The prototype shows the best performance when discriminating between AN events and non-AN events. The percentage of subbasins for which the prototype shows skill over HE in the upper tercile ranges between 85% and 98% for hindcasts initialised in the first three months ($S^2$ = 47-89%, and $S^1$ = 48-84%) then 57% and 25% for the last two months respectively ($S^2$ = 63% and 53%, and $S^1$ = 80% and 32%). The mean ΔROCSS ranges between -0.02 and 0.13 ($S^2$ = -0.01 and 0.14, and
$S^1$ = -0.01 and 0.07). The percentage of subbasins in cluster $S^3$ for which ΔROCSS > 0 is statistically significant are 18%, 10%, and 3% for hindcasts initialised in January, February, and March respectively, and 16% for forecasts initialised in January and February in cluster $S^2$. There are no statistically significant results in favour of HE.

**3.3 Analysis of the forecast ensemble sharpness**

Figure 6 shows the cross validated hindcasts by the prototype initialised in January (top panel) and May (bottom panel) for
Göuta-Ajaure, a cluster $S^3$ subbasin in upper reaches of the Ume River system. The total ensemble spread (the whiskers) of the forecasts initialised in January remains somewhat consistent from year to year while the IQR (the blue boxes) displays a more pronounced variation. The lack of variation in total spread is primarily the result of the climatological nature of the HE component which tends to have a larger and more consistent spread than that for DE and SE at longer lead times. The greater





variation exhibited by the IQR is mostly due to the 'true' forecast nature of the DE and SE components in the multi-model ensemble. If we turn our attention to the forecasts initialised in May we see a more pronounced variation in both the total spread and the IQR. This is because the spread in the DE and SE components is now comparable to and often larger than the spread in the HE component. Table 4 shows how the IQRSS drops as the spring flood season approaches. It can also be seen

in figure 6 that the ensemble median (red line) is more responsive to the year on year variation in SFV in the May forecasts than in the January forecasts. This is because the relative contribution to predictability by the initial conditions is greater than the contribution from the meteorological drivers closer to the spring flood period. These patterns are generally true for both the forecasts initialised in the intermediate months and for the other subbasins.

If we assume that the more sensitive an ensemble is to uncertainty the more the forecast sharpness will vary. We would

therefore expect the USS values to generally be positive i.e. that the forecast sharpness of prototype is better correlated with the forecast error than for HE. This is largely supported by the USS values in Table 4 where only three values are negative, the January forecast in cluster $S^2$ and the April forecasts in both clusters $S^1$ and $S^2$, and even then not by very much. This suggests that at least one but probably both of the DE and SE ensembles are responsible for this improvement due to their variability. There is a general decreasing trend with initialisation date in the USS values in clusters $S^1$ and $S^2$ (if we ignore

the value for January in $S^2$) while the values are more consistent in cluster $S^3$. All the uncertainty correlation values for both the HE and the prototype are significant at the 0.1 level (not shown for brevity) suggesting that both exhibit sensitivity to uncertainty to some extent, however prototype is generally more so which should instil more confidence for the forecast in the users.

The IQRSS values show that the prototype tends to produce sharper forecasts than HE early in the season but this reverses

itself in forecasts initialised in March for cluster $S^1$ and those initialised in April for the other clusters. This is probably due to the climatological nature of the HE having less of an impact on forecast sharpness as the initialisation date approaches the spring flood period together with the uncertainties and biases in the other individual ensembles exacerbating the situation.

### 4 Conclusions

In this paper we present the development and evaluation of a hydrological seasonal forecast system prototype for predicting

the SFV in Swedish rivers. Initially, two versions of the prototype, $ME_{ads}$ and $ME_{hds}$, were evaluated together with the HE using climatology as a reference to both help select which version of the prototype to proceed with and to get a general impression of their skill to forecast the SFV. Thereafter the chosen prototype was evaluated using HE as a reference and finally the sharpness of the hindcast ensembles were analysed.

Both multi-model ensembles show skill at forecasting SFV with respect to forecast error, ability to reproduce the interannual

variability in SFV, and the ability to discriminate between BN, NN, and AN events. At the least they have comparable skill to the HE when using climatology as a reference. Of the two proposed prototypes $ME_{hds}$ showed the most skill and was therefore chosen for the prototype. The prototype is shown to exhibit skill over the HE for most of the subbasins and





initialisation dates, although this skill is diminished when it comes to the discrimination between events. The prototype is at worst comparable to the HE and at best clearly more skilful. This means that for the user the prototype offers a seasonal forecast of the SFV that shows improved forecasting accuracy, better event prediction for early forecasts, and higher sensitivity to uncertainty. On average, over all subbasins and initialisation dates, the prototype is able to reduce the forecast

error by 6% and outperforms the HE 65% of the time. It is hoped that these improvements will make the forecasts more actionable for the users. The prototype was put into operation as a beta product at SMHI in January 2017.

Looking forward, future studies need to address the questions raised by Zhao et al. (2017) regarding the bias adjustment of meteorological seasonal forecast data using quantile mapping. Results from this study show that while the seasonal forecasts were bias adjusted the performance of the DE was disappointing, although it still had value within the multi-model setting

suggesting that it has more of a modulating roll on the other modelling chains as opposed to contributing directly to predictability.

The AE approach did not exhibit the promising performances found by Olsson et al. (2016) using circulation pattern analysis to select the analogues. A part of the explanation for this poor performance is that the teleconnection information used to select the analogues only partially span the full periods Foster et al. (2016) identified, from October/November to the

beginning of the spring flood. The missing data could be filled by making forecasts of the indices. Another approach would be to revisit the circulation pattern analysis based approach now that data inhomogeneity issues are largely addressed by the new ERA5 reanalysis data that is becoming available (http://climate.copernicus.eu/products/climate-reanalysis). Yet another approach would be to use GCM forecasts to select the analogues (e.g. Crochemore et al., 2016).

Lastly, the post processing of model outputs (e.g. Lucatero et al., 2017) has been shown to be beneficial, the incorporation of

a simple approach like linear scaling is possibly the most appealing due to its ease of implementation in an operational environment.

*Author contributions.* The AE and SE approaches were designed by K. Foster and C.B. Uvo and were implemented by K. Foster. The DE approach was designed by J. Olsson and implemented by K. Foster. The multi-model experimental set-up

was designed and implemented by K. Foster. The manuscript was prepared by K. Foster with contributions from C.B. Uvo and J. Olsson.

*Acknowledgements.* This work was supported by research projects funded by Energiforsk AB (formally Elforsk AB) and the EUPORIAS (EUropean Provision Of Regional Impacts Assessments on Seasonal and decadal timescales) project funded by

the Seventh Framework Programme for Research and Technological Development (FP7) of the European Union (grant agreement no. 308291). Many thanks go to Johan Södling, Jonas German, and Barbro Johansson for technical assistance as well as fruitful discussions.




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





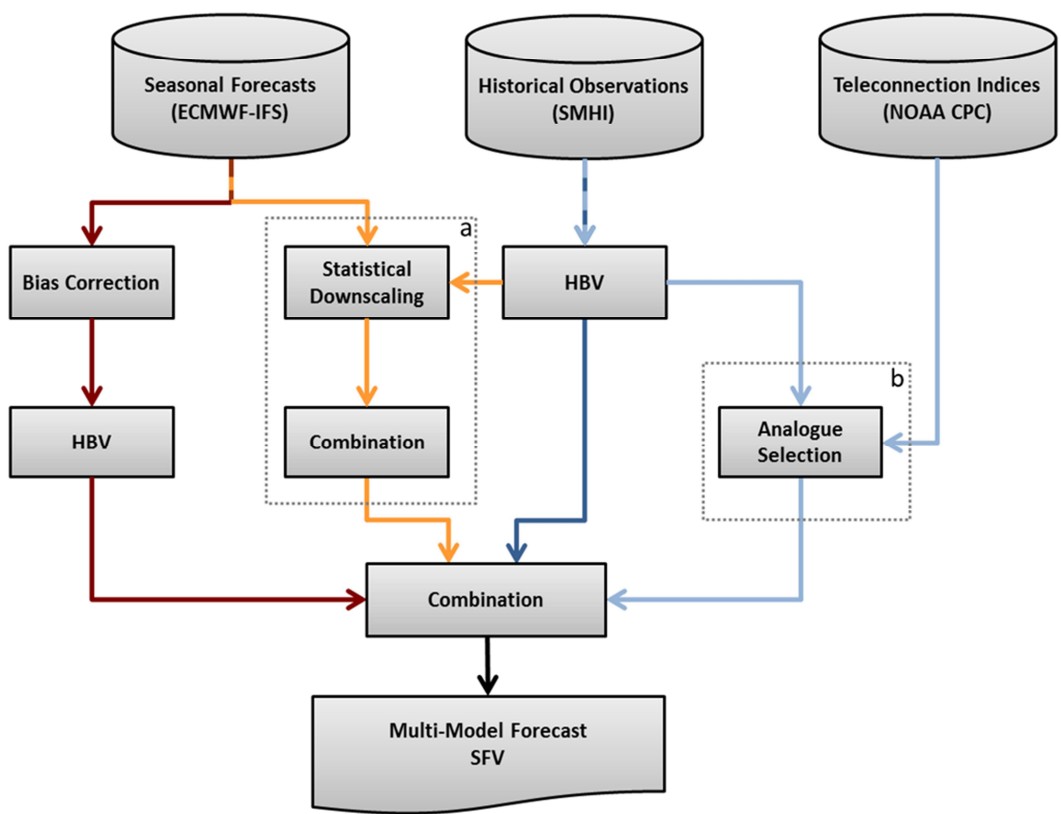

**Figure 1. Schematic of the multi-model forecast system. The three individual model chains that are included in the multi-model are (from left to right) the dynamic model chain (red lines), the statistical model chain (orange lines), and the historical (dark blue lines) or analogue (light blue lines) model chain. The dashed boxes labelled (a) and (b) indicate the parts of the system that have non trivial changes from the multi-model described in Olsson et al. (2016).**





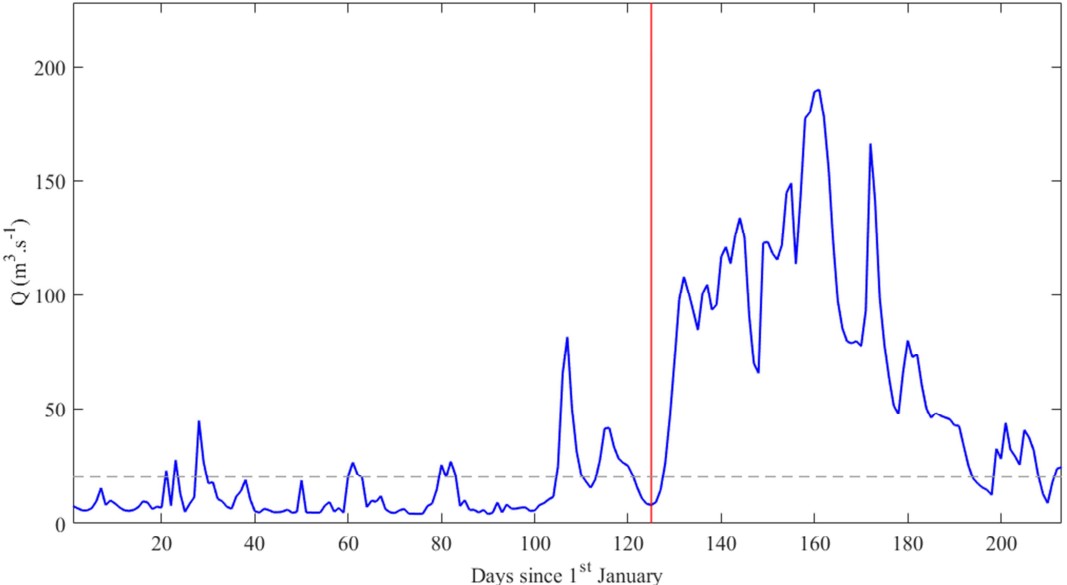

**Figure 2. Schematic of how the spring flood is defined. The spring flood is the period between the onset and the last day of July. The hydrograph from which the spring flood period is to be derived (blue line), the onset date (red line), and the 90th percentile of the inflow for first 80 days (dashed line).**





**Table 1. The validation metrics used to evaluate the multi-model performance. The threshold for skill is 50 for FY$^+$ and 0 for all the other metrics.**

| Name | Equation | Description |
|---|---|---|
| MAESS Mean absolute error skill score (**MAESS**) | $$MAESS = 1 - \frac{MAE_f}{MAE_r}$$ | Measure of the model's general performance; it quantifies the relative forecast error against a reference forecast. |
| Frequency of Years (**FY$^+$**) | $$FY^+ = \frac{100}{n}\sum_{y=1}^{n} H^y,$$ where $H$ is the Heaviside function defined by $$H^y = \begin{cases} 0, & AE_r^y < AE_f^y \\ 1, & AE_r^y > AE_f^y \end{cases},$$ $AE$ is the absolute error. | Measure of the model's general performance; it quantifies how often the forecast outperforms a reference forecast. |
| Nash-Sutcliffe efficiency (**NSE**) | $$NSE = 1 - \frac{\sum_{y=1}^{n}\left(SFV_{obs}^y - SFV^y\right)^2}{\sum_{y=1}^{n}\left(SFV_{obs}^y - \overline{SFV}_{obs}\right)^2}$$ | Measure of the model's general performance; it quantifies the model's residual variance against a reference forecast's variance. |
| Relative operating characteristic skill score (**ROCSS**) | $$ROCSS = 2 * AUC - 1,$$ where AUC is the area under the curve $$AUC = \sum_{y=1}^{n+1}\frac{(FR^y - FR^{y-1})(HR^y + HR^{y-1})}{2},$$ where $FR$ is the false alarm rate and $HR$ is the hit rate. | Measure of the model's probabilistic performance; it quantifies the model's ability of the discriminate between an event and a non-event given a specific threshold. |
| Interquartile range skill score (**IQRSS**) | $$IQRSS = 1 - \frac{IQR_f}{IQR_r}$$ where IQR is the interquartile range. | Measure of the forecast sharpness, it quantifies the relative spread in the forecast against a reference forecast. |
| Uncertainty sensitivity skill score (**USS**) | $$USS = \frac{(\rho_r - \rho_f)}{(1 - \rho_r)},$$ where $\rho$ is the Spearman rank correlation between the $IQR$ and absolute error. | Measure of the model's sensitivity to uncertainty; it quantifies the correlation between forecast sharpness and absolute error |





**Table 2. Basic information on the study area including overall performance of the HBV model for the subbasins in each cluster.**

| Cluster | | Basin | | SFV | | | HBV | |
|---|---|---|---|---|---|---|---|---|
| | | Area | elevation | (m³ x 10⁸) | | | NSE | rMAE |
| | | (km²) | (m) | min | mean | max | | (%) |
| | min | 233 | 135 | 0.21 | 0.42 | 0.82 | -0.47 | 3.9 |
| 1 | median | 1827 | 282 | 1.27 | 3.23 | 5.30 | 0.74 | 11.2 |
| | max | 6258 | 584 | 18.95 | 34.36 | 44.36 | 0.95 | 44.5 |
| | min | 184 | 429 | 0.40 | 0.81 | 1.68 | -0.69 | 6.2 |
| 2 | median | 1166 | 598 | 2.49 | 3.80 | 4.77 | 0.66 | 10.2 |
| | max | 4272 | 666 | 9.99 | 13.56 | 18.17 | 0.83 | 70.1 |
| | min | 270 | 212 | 0.67 | 1.12 | 1.54 | 0.22 | 3.8 |
| 3 | median | 1309 | 586 | 2.96 | 5.41 | 7.95 | 0.74 | 7.3 |
| | max | 13177 | 776 | 19.39 | 37.76 | 48.50 | 0.92 | 20.0 |





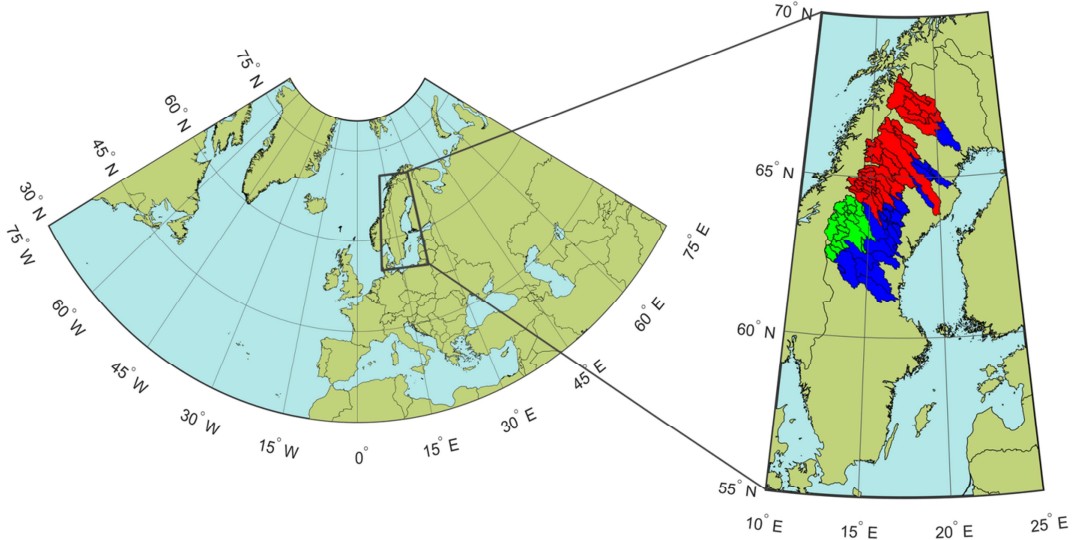

**Figure 3.** Map showing (left) the domain for the predictors used in the SE modelling chain, (right) the domain of the seasonal forecast data used in the DE modelling chain, and the location of the forecasts subbasins used in this work. The subbasins shown in blue belong to cluster $S^1$, those shown in green belong to cluster $S^2$, and those shown in red belong to cluster $S^3$.





**Table 3. Bootstrapped (N = 10000) skill scores and the number of subbasins, as a percentage, where the HEPS performs better than climatology averaged over all 84 subbasins. The $n^+$ values in brackets show the percentages of the subbasins for which these scores are statistically significant at the 0.1 level.**

| | | MAESS | $n^+$ (%) | NSE | $n^+$ (%) | ROCSS LT | $n^+$ (%) | MT | $n^+$ (%) | UT | $n^+$ (%) |
|---|---|---|---|---|---|---|---|---|---|---|---|
| | Jan | -0.09 | 25 (1) | -0.24 | 17 (0) | 0.23 | 90 (21) | 0.07 | 70 (0) | 0.10 | 68 (11) |
| | Feb | 0.00 | 51 (6) | -0.07 | 42 (5) | 0.41 | 99 (52) | 0.11 | 69 (1) | 0.26 | 92 (27) |
| HE | Mar | 0.09 | 80 (17) | 0.13 | 77 (23) | 0.55 | 100 (87) | 0.10 | 73 (5) | 0.44 | 99 (56) |
| | Apr | 0.15 | 85 (35) | 0.22 | 80 (35) | 0.62 | 100 (92) | 0.17 | 85 (7) | 0.51 | 100 (75) |
| | May | 0.21 | 90 (49) | 0.32 | 90 (49) | 0.68 | 100 (98) | 0.23 | 92 (10) | 0.61 | 100 (92) |
| | | | | | | | | | | | |
| | Jan | 0.00 | 50 (2) | 0.00 | 55 (1) | 0.31 | 99 (31) | -0.01 | 48 (0) | 0.20 | 80 (18) |
| | Feb | 0.06 | 73 (23) | 0.11 | 76 (21) | 0.39 | 99 (51) | 0.08 | 74 (0) | 0.36 | 96 (42) |
| $ME_{ads}$ | Mar | 0.11 | 86 (25) | 0.20 | 87 (36) | 0.47 | 100 (76) | 0.07 | 61 (4) | 0.47 | 100 (60) |
| | Apr | 0.20 | 95 (62) | 0.32 | 94 (64) | 0.60 | 100 (90) | 0.16 | 83 (5) | 0.52 | 100 (79) |
| | May | 0.22 | 96 (67) | 0.36 | 98 (68) | 0.66 | 100 (94) | 0.18 | 82 (8) | 0.57 | 100 (76) |
| | | | | | | | | | | | |
| | Jan | 0.02 | 60 (6) | 0.03 | 63 (5) | 0.32 | 100 (31) | 0.00 | 51 (0) | 0.22 | 83 (24) |
| | Feb | 0.08 | 80 (25) | 0.14 | 85 (29) | 0.41 | 99 (57) | 0.07 | 69 (1) | 0.38 | 99 (44) |
| $ME_{hds}$ | Mar | 0.14 | 90 (32) | 0.24 | 92 (45) | 0.51 | 100 (81) | 0.07 | 61 (5) | 0.48 | 100 (64) |
| | Apr | 0.19 | 94 (56) | 0.32 | 93 (62) | 0.60 | 100 (90) | 0.17 | 88 (5) | 0.54 | 100 (80) |
| | May | 0.24 | 98 (74) | 0.39 | 96 (76) | 0.67 | 100 (94) | 0.18 | 85 (10) | 0.60 | 100 (88) |





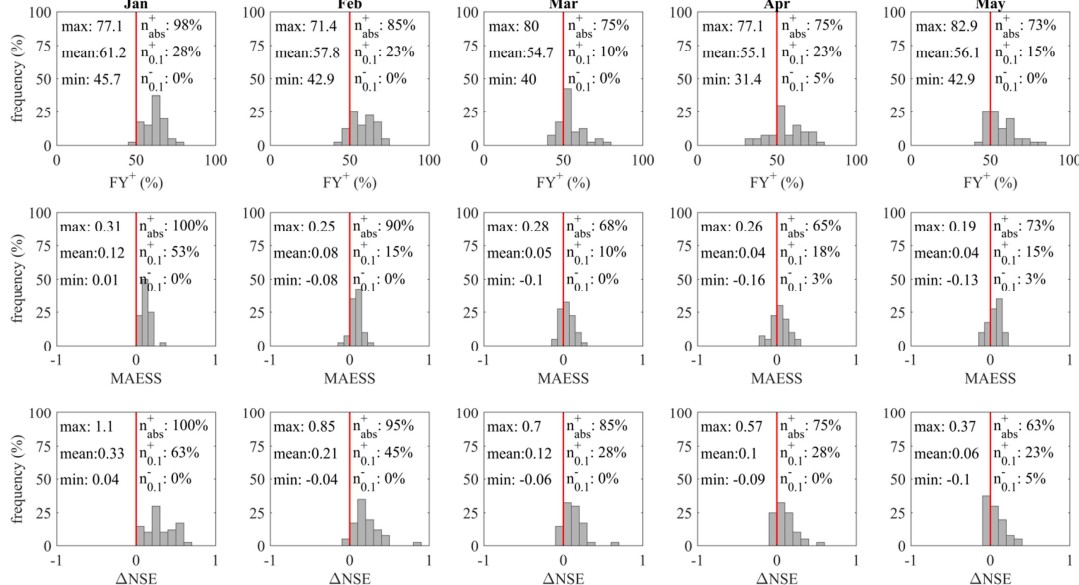

**Figure 4.** Bootstrapped (N = 10000) FY+, MAESS, and ΔNSE scores for $ME_{hds}$ with respect to HE for all subbasins in the cluster $S^3$. Each subplot is a histogram of the medians of the bootstrapped validations scores for each initialisation month. Above the histograms are six related statistics: (left of the red line) the maximum, mean, and minimum of the validation scores shown in the histograms; (right of the red line) percentages of the subbasins where $ME_{hds}$ performed better than HE ($n^+_{abs}$), the percentage of subbasins where $ME_{hds}$ performed better than HE ($n^+_{0.1}$) at the significance level 0.1, and lastly the percentage of subbasins where $ME_{hds}$ performed worse than HE ($n^-_{0.1}$) at the 0.1 level.





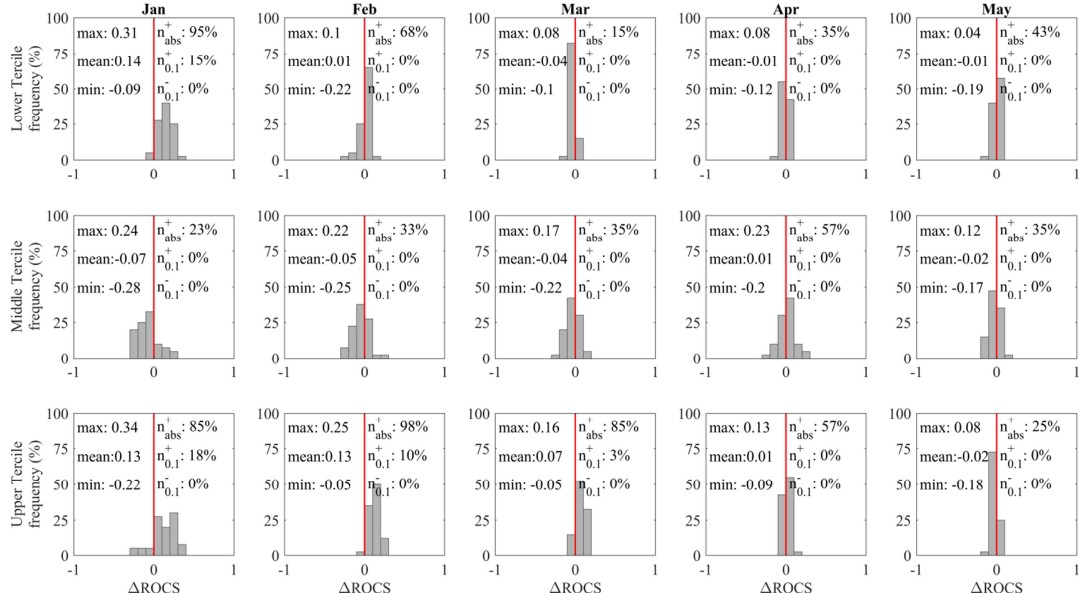

**Figure 5. Bootstrapped (N = 10000) ΔROCSS for the lower, middle, and upper terciles between the ME_hds and HE for subbasins in the cluster S³. Each subplot is a histogram of the medians of the bootstrapped validation score's ensembles for each initialisation month. Above the histograms are six related statistics: (left of the red line) the maximum, mean, and minimum of the validation scores shown in the histograms; (right of the red line) percentages of the subbasins where ME_hds performed better than HE ($n_{abs}^+$), the percentage of subbasins where ME_hds performed better than HE ($n_{0.1}^+$) at the significance level 0.1, and lastly the percentage of subbasins where ME_hds performed worse than HE ($n_{0.1}^-$) at the 0.1 level.**





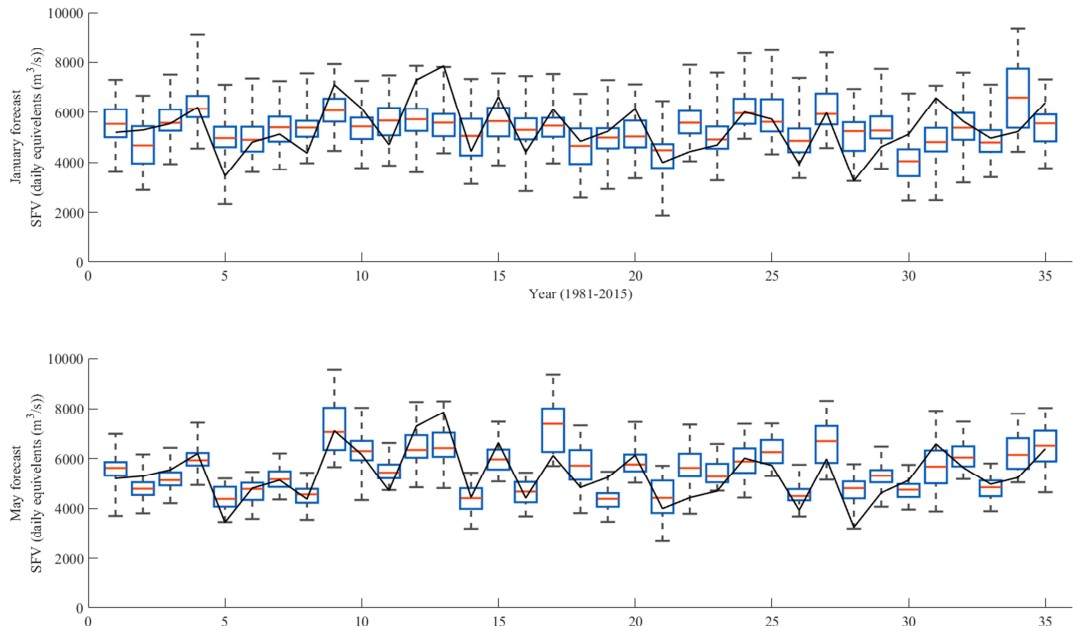

**Figure 6. The cross validated hindcasts of the SFV for a subbasin in cluster S³ made by ME_hds (boxplots) together with the observed SFV (black line). The box plots represent the entire forecast ensemble, the red lines represent the ensemble medians, the blue boxes the 25th and 75th percentiles (IQR), and the feelers represent the 0th and 100th percentiles.**





Table 4. Bootstrapped (N = 10000) USS and IQRSS for $ME_{hds}$ using HE as a reference. All values that are in bold are statistically significant at the 0.1 level.

| | | USS | | | | | IQRSS | | | | |
| --- | --- | --- | --- | --- | --- | --- | --- | --- | --- | --- | --- |
| | | Jan | Feb | Mar | Apr | May | Jan | Feb | Mar | Apr | May |
| $S^1$ | SS | 0.21 | 0.04 | 0.02 | -0.02 | 0.00 | 0.01 | 0.05 | -0.02 | -0.08 | -0.18 |
| | $n^+$ (%) | 80 (16) | 64 (0) | 56 (8) | 48 (4) | 52 (8) | 56 (24) | 72 (20) | 40 (8) | 16 (4) | 16 (4) |
| $S^2$ | SS | -0.05 | 0.17 | 0.07 | -0.10 | 0.17 | 0.01 | 0.06 | 0.05 | -0.13 | -0.15 |
| | $n^+$ (%) | 48 (4) | 80 (4) | 60 (4) | 40 (0) | 68 (4) | 52 (36) | 68 (40) | 64 (16) | 16 (4) | 12 (0) |
| $S^3$ | SS | 0.05 | 0.06 | 0.11 | 0.07 | 0.06 | 0.09 | 0.08 | 0.02 | -0.05 | -0.03 |
| | $n^+$ (%) | 60 (12) | 65 (10) | 65 (15) | 58 (10) | 60 (12) | 70 (52) | 80 (48) | 60 (25) | 32 (8) | 48 (18) |