# Peer review of "The development and evaluation of a hydrological seasonal forecast system prototype for predicting spring flood volumes in Swedish rivers"

_Hydrology and Earth System Sciences, 2017_

## Referee Comment (RC1) · J. Beckers (Referee) · 3 Nov 2017

**The development and evaluation of a hydrological seasonal forecast system prototype for predicting spring flood volumes in Swedish rivers**

Kean Foster, Cintia Bertacchi Uvo, Jonas Olsson

**General remarks**

This paper describes results from a hindcast study of a multimodel approach for ensemble streamflow forecasting. While the method has been described in an earlier paper, this paper focuses on the performance evaluation of two variations: MEads and MEhds. The skill metrics are well explained and results show a clear improvement of skill compared to the historical ensemble (HE, or classical ESP) method.

The text is well-written and clear. Below are a few minor comments and questions.

**Questions and comments**

- Page 3, line 30: It surprises me that no weighting scheme was applied. I would expect you have knowledge about the relative performance of the different model chains from experience of from earlier studies (Olson et al, 2016). Is this something you intend to investigate in the future?
- Page 4, line 29: Do you use all index values from this period, or the average, or else?
- Page 10, line 21 and 22: "starting from 1961" and "period 1961-2015" Is this a typo? On page 7, line 2, a different period was mentioned (1981-2015). If this is not a typo, why can the years 1961-1980 not be used for the performance evaluation?

**Grammar and spelling**

- Use punctuation when using adjuncts, for example on page 2:
    - To achieve this, operators …
    - In practice, there are …
- Page 10, line 6: " sub-basins"
- Page 10, line 17: "( SFV)" (this abbreviation has been introduced before).
- Page 12, line 20: "… to perform  better than…"
- Page 13, line 16: " mean"
- Page 14, line 17: "with regard to"
- Page 15, line 17: "however the prototype"
- Table 3: LT, MT and UT are not introduced. Do you mean BN, NN, AN?

---

## Referee Comment (RC2) · F. Mainardi Fan (Referee) · 8 Nov 2017

Brief Overview The paper presents a very interesting study related to the implementation of a prototype for seasonal forecasting in Swedish rivers based on hydrological modelling and seasonal meteorological forecasts. The prototype is compared to a traditional operational EPS approach and to climatology. Results show benefits in the use of the prototype. The paper is well written, methods are adequately described, and assessments seems suitable to the objectives. I have only a few major and minor comments about the manuscript.

Major Comments P2, l20-25: Please observe that it historical observations are referred two times. And only in the second one it is presented as the ESP approach. The explanation here could be better. Evaluation section: I understand that one of the limitations of the work is that authors were not able to evaluate properly the ensemble, since most of the used metrics are related to transforming the ensemble into the ensemble mean, and then evaluating it as a deterministic forecast. Authors did not even experiment testing some other metrics? P5, l10: It is relevant to better explain what is the data used in the bias correction. Also, I think this procedure has great impact in results, but it is not adequately described. My suggestion is to explore more this point. Conclusions: Authors commented that the prototype was put into operation as a beta product at SMHI in January 2017. This gives openness for another discussion: in an operational perspective, are the benefits verified for the prototype enough to justify the implementation? I understand that yes, but also the prototype is more dependent on data and require more processing power and time to run, right?

Minor Comments P1, l16: "considered" is doubled in the text; P2, l34: Please explain better what is "limited success". Only one case is cited; P15, l19: The sentence is confusing. Please revise. P6, l10: "subbasins sub-basins"

---

## Author Comment (AC1) · 14 Nov 2017

**Response to reviewer 1**

Reviewer's comments are in blue, our comments are in black.

**General remarks**

This paper describes results from a hindcast study of a multimodel approach for ensemble streamflow forecasting. While the method has been described in an earlier paper, this paper focuses on the performance evaluation of two variations: MEads and MEhds. The skill metrics are well explained and results show a clear improvement of skill compared to the historical ensemble (HE, or classical ESP) method.

The text is well-written and clear. Below are a few minor comments and questions.

**Questions and comments**

Page 3, line 30: It surprises me that no weighting scheme was applied. I would expect you have knowledge about the relative performance of the different model chains from experience of from earlier studies (Olson et al, 2016). Is this something you intend to investigate in the future?

Yes, we understand your surprise here. The wording used (page 4, lines 1-2) implies we did not test weighting schemes which is not the case. We did in fact test two types of weighting, a simple arithmetic weighting system similar to that used by Olsson et al. (2016) and a linear regression based approach. In both cases the sharpness of the multi-model ensemble was improved significantly but there was no improvement in the multi-model's general performance. The pooling approach was still able to hold a small advantage over the weighted versions.

The following changes were made to correct and clarify what we did:

page 4, lines 1-2 was changed to read,

"The simple weighting scheme used by Olsson et al. (2016) was tested but, other than improving the ensemble sharpness, did not offer an improvement over the pooling approach."

The following paragraph has been added to the conclusion as well:

"How the individual model ensembles are combined to give the multi-model output needs to be revisited. When we applied the asymmetric weighting scheme proposed by Olsson et al. (2016) we did not find that it improved the multi-model performance in general across all stations and forecasts and so did not use it. However, we do believe that more work should be done to find a more appropriate weighting scheme than simple pooling. Perhaps by better understanding how the performance of the different modelling chains are affected by the initial conditions and lead-time it will shed more light on how to best approach this issue. Further development and testing along these lines are planned for the future."

Page 4, line 29: Do you use all index values from this period, or the average, or else?

We use the mean or average index values for the period. We have reworded the sentences (page 4, 29) to clarify this,

"The teleconnection indices they identified are the Arctic oscillation (AO) and the Scandinavian pattern (SCA) and the periods of persistence for these indices, expressed as the index mean for the identified period, are the seven and eight months leading up to the spring flood respectively."

Page 10, line 21 and 22: "starting from 1961" and "period 1961-2015" Is this a typo? On page 7, line 2, a different period was mentioned (1981-2015). If this is not a typo, why can the years 1961-1980 not be used for the performance evaluation?

This is a typo. The reason that the entire data series cannot be used is that the hindcasts of the driving data used in the multi-model are only available from 1981.

We have changed it to read, "…the data used in our work are for the period 1981-2015 due to some of the other  datasets used in this work only being available from 1981".

**Grammar and spelling**

Use punctuation when using adjuncts, for example on page 2:

- To achieve this, operators …
- In practice, there are …

The text has been reviewed and punctuation added when using adjuncts as suggested.

The second hyphenated word has been removed.

Page 10, line 17: "(hereafter SFV)" (this abbreviation has been introduced before).

Changed to read, "We focus on forecasts of the accumulated streamflow volume during this period or SFV."

The word 'the' has been removed so that it now reads, "… to perform better than…"

The 'an' has been changed to an 'a' to read, "a mean".

The phrase has been replaced with the 'at' so the sentence reads, "Out of the three terciles the prototype shows the least skill over HE at discriminating between NN events and non-NN events."

The word 'the' has been added as suggested.

Table 3: LT, MT and UT are not introduced. Do you mean BN, NN, AN?

Yes, this is a typo and we did mean BN, NN, AN. The change has been made.

---

## Author Comment (AC2) · 14 Nov 2017

**Response to reviewer 2**

Reviewer's comments are in blue, our comments are in black.

**Brief Overview**

The paper presents a very interesting study related to the implementation of a prototype for seasonal forecasting in Swedish rivers based on hydrological modelling and seasonal meteorological forecasts. The prototype is compared to a traditional operational EPS approach and to climatology. Results show benefits in the use of the prototype. The paper is well written, methods are adequately described, and assessments seems suitable to the objectives. I have only a few major and minor comments about the manuscript.

**Major Comments**

P2, l20-25: Please observe that it historical observations are referred two times. And only in the second one it is presented as the ESP approach. The explanation here could be better.

We have reworded page 2, lines 19 and line24, to make this clearer. They now read as follows:

"…and then force it with either historical observations (called ensemble streamflow prediction or ESP; e.g. Day, 1985)…"

"Another dynamical approach is the well-established ESP method (Day, 1985)."

Evaluation section: I understand that one of the limitations of the work is that authors were not able to evaluate properly the ensemble, since most of the used metrics are related to transforming the ensemble into the ensemble mean, and then evaluating it as a deterministic forecast. Authors did not even experiment testing some other metrics?

With only 35 data points per station, one data point per year, we felt that it was not enough data on which to perform a robust probabilistic evaluation on. We experimented with the metric CRPS but were ultimately uncomfortable presenting those results due to their uncertainty arising from the limited data used in the analysis. We should also point out that the inter quartile range skill score (IQRSS) and uncertainty sensitivity skill score (USS) used in this work are basic ensemble evaluation metrics and, although not a full probabilistic evaluation, do give some insight into the performance of the forecast ensembles.

P5, l10: It is relevant to better explain what is the data used in the bias correction. Also, I think this procedure has great impact in results, but it is not adequately described. My suggestion is to explore more this point.

We have expanded our description of the bias adjustment and have replaced page 5, line 10 with the following:

"A change to previous work has these daily P and T data bias adjusted first before being used to force HBV. The bias adjustment method used is a version of the distribution based scaling approach (DBS; Yang et al., 2010) which has been adapted for use on seasonal forecast data. DBS is a quantile mapping bias adjustment method where meteorological variables are fitted to appropriate parametric distributions (e.g. Berg et al., 2015; Yang et al., 2010). For precipitation, two discrete gamma distributions are used to adjust the daily seasonal forecast values, one for low-intensity precipitation events (≤ 95th percentile) and another for extreme events (> 95th percentile). For temperature, a Gaussian distribution is used to adjust the daily seasonal forecast values.

Observed (Sect. 2.6 Study area and local data) and seasonal forecast (Sect. 2.7 Driving Data) time-series of P and T spanning the relevant forecast timeframe (e.g. Jan-Jul for forecasts initialised in January) and for the reference period 1981-2010 are used to derive the adjustment factors to transform the seasonal forecast data to match the observed frequency distributions. First the precipitation data is adjusted then the temperature data. The latter is done separately for dry and wet days in an attempt to preserve the dependence between P and T (e.g. Olsson et al. 2010; Yang et al, 2010). Adjustment factors are calculated for each calendar month as the distributions can have different shapes depending on the physical characteristics of the precipitation processes that are dominant. It should be emphasized that the adjustment parameters were estimated using much of the same data to which they were applied. Ideally the parameters would be estimated using data that does not overlap the data which is being adjusted. However, this was not possible in the scope of this work."

**Conclusions:**

Authors commented that the prototype was put into operation as a beta product at SMHI in January 2017. This gives openness for another discussion: in an operational perspective, are the benefits verified for the prototype enough to justify the implementation? I understand that yes, but also the prototype is more dependent on data and require more processing power and time to run, right?

Yes, we and the power companies think that they do. It must be emphasised that every percent improvement in the forecast error can potentially be converted into large financial revenues for the power companies and energy traders. So an average improvement in forecast error, over all subbasins and initialisation dates, by 6% (individual results can be as high as 31%, see figure 4) can be viewed as a significant improvement. Care was taken while developing the prototype to minimise the added computational power and data requirements. Additionally, these forecasts are made only once a month so the additional computational time, ca. 1 extra hour, is not a significant factor.

Page 16, line 6 was rewritten to emphasise that the implementation of the prototype as a beta product was done together with the power companies. It now reads:

"These results have been met with great interest from the hydropower industry and the prototype was put into operation, in cooperation with the power companies, as a beta product at SMHI in January 2017."

**Minor Comments**

*P1, l16: "considered" is doubled in the text*

The first instance has been deleted so that it now reads, "Both the considered multi-model methods considered showed skill over the reference forecasts…"

*P2, l34: Please explain better what is "limited success". Only one case is cited*

The paper cited is a review of the different experiments performed at SMHI to improve the forecast error of the SFV. They found that a despite these efforts the

*P15, l19: The sentence is confusing. Please revise.*

The line has been reworded to read, "The IQRSS values show that the prototype tends to produce sharper forecasts than HE early in the season i.e. for forecasts initialised in January and February in cluster $S^1$ and forecasts initialised in January, February and March in clusters $S^2$ and $S^3$. This is reversed for the remaining initialisation dates where HE tends to produce sharper forecasts than the prototype."

*P6, l10: "subbasins sub-basins"*

The second hyphenated instance has been deleted.

---

## Short Comment (SC1) · 30 Nov 2017

Dear authors and editor,

In the "Current topics in Earth System Science" (ESS 401) course at the University of Zurich, we ask students to choose a manuscript that was recently submitted to one of the Copernicus journals and to write a review for this manuscript. Two students (Hannes Tobler and Sebastian Röthlin) chose your manuscript and wrote the attached review. We hope that you find it interesting and useful.

Best regards,

[Figure]

Ilja van Meerveld

Please also note the supplement to this comment:
https://www.hydrol-earth-syst-sci-discuss.net/hess-2017-588/hess-2017-588-SC1-
supplement.pdf
588, 2017.

[Figure]

**Supplement:**

**Review on: The development and evaluation of a hydrological seasonal forecast system prototype for predicting spring flood volumes in Swedish rivers**

**Summary of the manuscript**

The manuscript shows, how a hydrological seasonal forecast system prototype is adjusted from the previous version and evaluated on its ability to predict spring flood volumes in Swedish rivers. The aim is to improve water resource management for hydropower decision makers. The study area consists of 84 subbasins in northern Sweden, which have a runoff regime that is strongly influenced by spring snow melt. The skill of the multi-model prototype is compared in cross-validated hindcasts to the historical ensemble streamflow prediction based on measurements between 1981 and 2015. This historical ensemble represents the setup currently used for hydropower reservoir management. The multi-model prototype represents combinations of the historical ensemble, an analogue ensemble (subset of the historical ensemble based on similarities in parameters of interannual climate variability), a dynamic modelling ensemble (bias-corrected seasonal forecast) and a statistical modelling ensemble (downscaled seasonal forecast). Several complementary, statistical measures were used for the evaluation of the new prototype. The prototypes, that combine 3 different ensembles show at best a significant improvement compared to the currently used historical ensemble and at worst a comparable skill.

**Main assessment**

Based on our assessment, the reviewed manuscript reaches a substantial conclusion based on sufficient results which generally were outlined clearly and used valid assumptions. Overall, we like the clear structure of the paper and the scientific notation. In the abstract and the introduction, it is nicely explained why there is a public benefit behind this research.

When we first read the introduction, we had some problems to understand the differences between the two approaches (dynamical and statistical). When we came to the points where it is explained better it is not a problem anymore. Maybe a quick hint to the section 2.14 and 2.15 would help the readers - or provide some of the clarifications already in the introduction.

It would have been useful to add the research questions to the introduction. The paper describes what has been done, how it was tested and how good the new prototype is for catchments in Sweden but not explicitly what science question or hypothesis is answered. The paper also does not discuss how applicable this method/prototype is for other snow dominated areas.

Thanks to the SMHI, the datasets are very good, but we don't understand why the subbasins with 61, 63, and 73 percent of missing data are also included in these 84 basins that are investigated. Has it at least been checked whether and how the results change when those three subbasins are excluded?

The evaluation part from this study is very elaborate and we like this. However, it could have been more clearly explained what a 6% improvement means, for example in volume of water or economic value for hydropower generation. Is this 6% really significant?

We would have appreciated it if the paper provided a little more information about the seasonal meteorological forecasts that are used. For example something about the uncertainty of these models or why the ECMWF IFS system is used (is there no other meteorological forecast system for six months or is it the best seasonal forecast system)?

Finally, it would have been interesting if the differences for the different catchments would have been discussed. Were the improvements mainly seen for large/small catchments, for high elevation/flat catchments? Some maps would have been nice as well.

**List of major and minor points**

Page one:

Lines 8-10: In our opinion a very catchy and smart opening.

Line 15 to 18: This sentence is maybe too long and little too complicated for the abstract. (Full stop before 'however'?). Twice 'considered'

Line 23: Unclear reference, presumably 'Statistiska centralbyrån'? For clarity, the abbreviation can be included in the reference (Statistiska centralbyrån (SCB): …)

Line 27 to 28: The idea or point behind the sentence comes across. But if you first read this sentence, it could be puzzling. We also think that with all these brackets the text looks not as nice as it could. Why not just add 'and vice versa' at the end of the sentence?

Page two:

Line 3: Grammatical: The strategy is to have reservoirs which are then managed.

Line 4: comma: To achieve this, operators …

Line 6: The meaning of the expression 'sources of predictability' is unclear to us in this situation. Can you explain briefly?

Line 7: Decide on stores within or in the catchment. We suggest using within.

Line 11: Probably : instead of ; after 'forecasts at the seasonal scale'.

Line 15: There is no need for a comma after the closed bracket. Both times.

Lines 19, 20: The second time 'force it' (end of line 19) is not necessary.

Lines 31, 32: Do the references explain how and to what extent historical observations of precipitation and temperature are possible representations of future meteorological conditions in the context of the ongoing climate change? Do the historical data show any significant trend?

Page three:

Line 13: Maybe already mention here the number of catchments and data period. That way we already know something about how these modeling steps are applied. Otherwise the 35 in Line 35 is not so clear.

Line 14 to 15: Twice the word "brief"

Line 19: first improved by Foster et al. (2010)  later improved  and first tested by Olsson et al. (2016).

Lines 23, 24: Here the manuscript includes already some results but is in the Materials and Methods section. Finish sentence after '… of these four were tested.'

Line 28: Replace 'relevant' by 'respective'.

Page four:

Line 7: Is there a reference on what the seasonal forecasting practice at SMHI is?

Line 9: It is not clear to us how the DBS method is different from the previous method.

Line 26: We are not familiar with the teleconnection approach. A brief explanation would have been useful.

Page five:

Line 2: What is the meaning of and the justification for a distance of 0.2?

Line 29 and following: Is there a reference (needed) for the physical support of the asymmetric weighting?

Page six:

Line 24: Typo: overfitted

Line 28: So n = 35 in this case?

Page seven:

Line 27: Why is the relative mean absolute error used (error divided by $SFV_0^y$)? Would the absolute error (without the division) not be a stronger focus on the flood peaks and therefore more beneficial for the assessment of the skill of forecasting the spring flood? Please explain.

Line 28: The superscripts suggest that it is SFV to the power of y. A different notation is clearer.

Page ten:

Line 6: Subbasin.

Line 6: There should be a reference to figure 3 included in the sentence.

Line 10: How is total runoff divided between the 3 subbasins? Would it be interesting to use the new setup also for the two other subbasins?

Line 17: SFV already introduced on page 3.

Line 27: We don't understand why the subbasins with 61, 63, and 73 percent of missing data are also included in the 84 basins that are investigated. Has it been checked whether and how the results change when excluding those three subbasins?

Line 31: What is the PTHBV dataset from SMHI? is this the ptq file for HBV (but without the q)?

Page eleven:

Line 4: Do you refer to the performance measures for the HBV rainfall-runoff model of the historical data?

Line 7: The 'than' near the end of the line is missing one or two adjectives describing latitude and elevation. Decide on either and or or.

Line 22-24: We don't really understand where the 15 and 51 come from. We probably missed that somewhere, or is it not explained anywhere?

Page thirteen:

Line 21: Are these 5% of the catchments located in a certain area or do they have similar characteristics? Large/small, steep/flat, northern/southern?

Page fourteen:

Line 29: Why was exactly this site choose for the analysis of the forecast ensemble sharpness? Just one sentence, why exactly this basin is relevant out of the whole set of the 84 sites.

Page sixteen:

Line 5-18: This is more discussion than real conclusion. Move?

Figures:

Figure 2: In the figure description it is mentioned that the spring flood is in the period between the onset and the last day of July. On the x-axis the days since the 1st January are written. We think that for the reader it would be easier if on the x-axis the first of the months were labelled and the date of the 80 days threshold was indicated (20th or 21st of March).

---

## Author Comment (AC3) · 2 Feb 2018

**Response to reviewer's comments in SC1**

Reviewer's comments are in blue, our comments are in black.

**Summary of the manuscript**

The manuscript shows, how a hydrological seasonal forecast system prototype is adjusted from the previous version and evaluated on its ability to predict spring flood volumes in Swedish rivers. The aim is to improve water resource management for hydropower decision makers. The study area consists of 84 subbasins in northern Sweden, which have a runoff regime that is strongly influenced by spring snow melt. The skill of the multi- model prototype is compared in cross-validated hindcasts to the historical ensemble streamflow prediction based on measurements between 1981 and 2015. This historical ensemble represents the setup currently used for hydropower reservoir management. The multi-model prototype represents combinations of the historical ensemble, an analogue ensemble (subset of the historical ensemble based on similarities in parameters of interannual climate variability), a dynamic modelling ensemble (bias-corrected season- al forecast) and a statistical modelling ensemble (downscaled seasonal forecast). Several complementary, statistical measures were used for the evaluation of the new proto- type. The prototypes, that combine 3 different ensembles show at best a significant improvement compared to the currently used historical ensemble and at worst a comparable skill.

**Main assessment**

Based on our assessment, the reviewed manuscript reaches a substantial conclusion based on sufficient results which generally were outlined clearly and used valid assumptions. Overall, we like the clear structure of the paper and the scientific notation. In the abstract and the introduction, it is nicely explained why there is a public benefit behind this research.

When we first read the introduction, we had some problems to understand the differences between the two approaches (dynamical and statistical). When we came to the points where it is explained better it is not a problem anymore. Maybe a quick hint to the section 2.14 and 2.15 would help the readers - or provide some of the clarifications already in the introduction.

**1)**

We have added a cross reference to page 2, line 11 and line 12. It now reads as follows:

"In practice, there are two predominant approaches to making hydrological forecasts at the seasonal scale; statistical approaches and dynamical approaches (see Sect. 2.1.4 and Sect. 2.1.5 for more regarding these approaches in the context of this work)."

It would have been useful to add the research questions to the introduction. The paper describes what has been done, how it was tested and how good the new prototype is for catchments in Sweden but not explicitly what science question or hypothesis is answered.

**2)**

This is a valid point. We have added a sentence after page 1, line 13 and line 14 to include what the overarching hypothesis of this work is. It reads:

"The hypothesis explored in this work is that a multi-model seasonal forecast system which incorporates different modelling approaches is generally more skilful at forecasting the SFV in snow dominated regions than a forecast system that utilises only one approach."

This, together with the beginning of the paragraph, now gives the reader a brief description of the issue this work is intended to address and the hypothesis that is tested.

The paper also does not discuss how applicable this method/prototype is for other snow dominated areas.

**3)**

We have added a paragraph that discusses this to the results and discussion section. See our response (no.7) to your comments:

Thanks to the SMHI, the datasets are very good, but we don't understand why the subbasins with 61, 63, and 73 percent of missing data are also included in these 84 basins that are investigated. Has it at least been checked whether and how the results change when those three subbasins are excluded?

**4)**

These stations were included because they are part of the operational forecast and are therefore relevant to the prototype. This we mention on page 11, line 25 and line 26. We did check how their inclusion affected the results and found that there was a small improvement in the skill of the prototype. This we attribute to an adverse affect they have on the statistical branch which needs to be trained on these historical data. However, the drop in skill is not such that it detracts from the general results. So, due to their importance to the prototype and the relative low impact hey have on the performance we chose to retain them in this work.

The evaluation part from this study is very elaborate and we like this. However, it could have been more clearly explained what a 6% improvement means, for example in volume of water or economic value for hydropower generation. Is this 6% really significant?

**5)**

Please see our response ($4^{th}$) to reviewer 2. Yes it is significant to the power companies. It is difficult to put clear contextual examples of what this improvements means economically.

We have made changes to the first third of the conclusion section (see our response no. 41 to your comments). In this we have included a volumetric interpretation of what a 6% reduction in SFV entails.  The bullet point reads:

"•      The prototype is able to reduce the forecast error by 6% on average. This translates to an average volume of 9.5 x $10^6$ $m^3$."

We would have appreciated it if the paper provided a little more information about the seasonal meteorological forecasts that are used. For example something about the uncertainty of these models or why the ECMWF IFS system is used (is there no other meteorological forecast system for six months or is it the best seasonal forecast system)?

**6)**

The choice to use the ECMWF as a provider of the seasonal forecasts to force the prototype is primarily based on operational considerations. SMHI has operational access to these products so no extra effort is needed to source and collect these data. It should be noted that there are other seasonal forecast data providers and we are looking to test them in the future, however that is not part of the scope of this work.

We have added two sentences after the line on page 11, line 20 briefly motivating our choice. They read:

"The choice to use ECMWF data is primarily a practical one. The ECMWF is an established and proven producer of medium range forecasts and SMHI already has operational access to their products."

Finally, it would have been interesting if the differences for the different catchments would have been discussed. Were the improvements mainly seen for large/small catchments, for high elevation/flat catchments? Some maps would have been nice as well.

**7)**

We have added a section to the Results and discussion section which addresses this point. It reads:

"3.4 Spatial and temporal variations and transferability of the prototype

Both multi-model ensembles show skill at forecasting SFV with respect to forecast error, ability to reproduce the interannual variability in SFV, and the ability to discriminate between BN, NN, and AN events. The prototype, in particular, is at worst comparable to the

HE and at best clearly more skilful. This relative performance of the prototype varies both in space and time. Figure 7 shows maps of the median bootstrapped FY+ values. For hindcasts initialised in January the spatial pattern in the FY+ scores show that the prototype tends to outperform HE more in subbasins that have a higher latitude or elevation. However, as the initialisation date approaches the spring flood period this pattern becomes less and less coherent. This general pattern is also true for MAESS scores. This suggests that the change in the performances of the prototype and HE, as a function of initialisation date, are not always similar for subbasins that are near one another. Further work would be needed to find out what the underlying reason for this is.

Data availability is the biggest limiting factor to the transferability of this approach to other areas. The HE, AE, and SE approaches are all dependant on good quality observation time-series. Additionally, the skill all three of these approaches would be expected to be affected by length of these time-series. They length of the time-series should be long enough to be a good representative sample of the climatology otherwise the forecasts would be biased in favour of the climate represented in the data and not the true climatology.

The SE and AE approaches require an understanding of how the variability in the local hydrology is affected by large scale circulation phenomena such as teleconnection patterns to help select predictors and teleconnection indices for inputs to each approach respectively. The hydrological rainfall-runoff model used in the prototype should not pose a problem, although HBV has been successfully setup for snow dominated catchments outside of Sweden (e.g. Seibert et al., 2010; Okkonen and Kløve, 2011), any sufficiently well calibrated rainfall-runoff model would suffice.

We believe that, if the above requirements are met, a seasonal hydrological forecast system similar to the prototype can be setup in other snow dominated regions around the world."

[Figure]

**Figure 7. Maps of the median bootstrapped FY+ values for each of the initialisation dates.**

List of major and minor points

Page one:

Lines 8-10: In our opinion a very catchy and smart opening.

**8)**

Thank you.

Line 15 to 18: This sentence is maybe too long and little too complicated for the abstract. (Full stop before 'however'?). Twice 'considered'

**9)**

This sentence has been broken up and now reads:

"Both the multi-model methods considered showed skill over the reference forecasts. The version that combined the historical modelling chain, dynamical modelling chain, and statistical modelling chain performed better than the other and was chosen for the prototype."

Line 23: Unclear reference, presumably 'Statistiska centralbyrån'? For clarity, the abbreviation can be included in the reference (Statistiska centralbyrån (SCB): …)

**10)**

The reference in page 1, line 23 has been changed from the abbreviation to the full reference.

Line 27 to 28: The idea or point behind the sentence comes across. But if you first read this sentence, it could be puzzling. We also think that with all these brackets the text looks not as nice as it could. Why not just add 'and vice versa' at the end of the sentence?

**11)**

Yes the use of vice versa does improve the readability of the sentence without detracting from the message. The sentence has been rewritten as suggested. It now reads:

"This reservoir management is important as the energy demand is out of phase with the natural availability of the water resources; typically demand is higher during the colder months when the inflows are lower and vice versa."

Page two:

Line 3: Grammatical: The strategy is to have reservoirs which are then managed. Line 4: comma: To achieve this, operators …

**12)**

Please see our response (4[th]) to reviewer 1.

Line 6: The meaning of the expression 'sources of predictability' is unclear to us in this situation. Can you explain briefly?

**13)**

The expression refers to where the signal that gives skill to the forecasts originates from. The SFV is a function of many hydrometeorological factors but some influence the variability of the SFV more than others. For example, in the context of this work, the snowpack is a major contributor to the SFV and therefor data related the amount of water stored in the

snowpack can potentially be used to make a skilful forecast. In this example, information regarding the snow pack is leading source of predictability in seasonal forecasts of the SFV in these regions.

**14)**

Line 7: Decide on stores within or in the catchment. We suggest using within. Line 11: Probably : instead of ; after 'forecasts at the seasonal scale'.

Yes, the former is a typo and the redundant word 'in' has been deleted. The latter suggestion of using a colon instead of a semi-colon has also been applied.

Line 15: There is no need for a comma after the closed bracket. Both times. Lines 19, 20: The second time 'force it' (end of line 19) is not necessary.

**15)**

These changes have been applied.

Lines 31, 32: Do the references explain how and to what extent historical observations of precipitation and temperature are possible representations of future meteorological conditions in the context of the ongoing climate change? Do the historical data show any significant trend?

**16)**

No they do not. The standard ESP approach assumes stationarity and does therefore not take into account changes in climate. Yes, there is a change signal in the historical data but making allowances for this was not within the scope of this work. However, there are future plans to investigate the added value of adjusting the historical data to mitigate this change signal before use in the modelling chain. Another approach, which we mention in the manuscript, is the post-processing of the forecasts to account for any biases related to factors such as climate change signals.

Page three:

Line 13: Maybe already mention here the number of catchments and data period. That way we already know something about how these modeling steps are applied. Other- wise the 35 in Line 35 is not so clear.

**17)**

We feel that by adding this information in page 3, line 13 would make the sentence clumsy to read. Instead, we added information on the number of catchments to page 3, line 21; we added information regarding the data period to page 3, line 30. The affected sentences now read:

"The aim is to adapt their methodology for use in an operational environment and then evaluate the resulting prototype against the current operational system using cross-validated hindcasts for 84 gauging stations in northern Sweden (see sect. 2.6)."

And

"These outputs are pooled together rather than using an asymmetric weighting scheme due to the lack of data points, a total of 35 spring flood events (hindcast period was 1981-2015, see Sect. 2.6), from which to derive a robust weighting scheme."

Line 14 to 15: Twice the word "brief"

**18)**

The second occurrence of brief has been removed

Line 19: first improved by Foster et al. (2010) and, later improved upon and first tested by Olsson et al. (2016).

**19)**

A comma was added.

Lines 23, 24: Here the manuscript includes already some results but is in the Materials and Methods section. Finish sentence after '… of these four were tested.'

**20)**

The sentence was shortened accordingly.

Line 28: Replace 'relevant' by 'respective'.

**21)**

The replacement was made.

Page four:

Line 7: Is there a reference on what the seasonal forecasting practice at SMHI is? Line 9: It is not clear to us how the DBS method is different from the previous method.

**22)**

We assume you are referring to page 5, line 7. Please see our 3[rd] response to reviewer 2.

Line 26: We are not familiar with the teleconnection approach. A brief explanation would have been useful.

**23)**

There is a brief explanation of the revised teleconnection approach later in the next paragraph (page 4, line 31 – page 5, line 5) which gives an overview of what the approach entails. However, we have changed the word 'the' to 'their' (page 4, line 26) to clarify that we are referring to an approach proposed by Olsson et al. (2016) which we have already referred to.

Line 2: What is the meaning of and the justification for a distance of 0.2?

**24)**

We are using the persistence in the teleconnection indices leading up to the forecast date to select analogue years out of the historical dataset. In order to be able to identify which of the historical years are analogues we need a selection criteria. We define an analogue to be any year whose Euclidean distance is less than 0.2 units from the Euclidean position of the 'current' year (see page 5, line1 – line2). The threshold is a compromise between being small enough to be sufficiently specific and being large enough to actually be able to capture some analogues from the historical data.

We appreciate that this is not entirely clear in the manuscript. We added to page 5, line 2 to clarify what the value 0.2 referred to. It now reads:

"If the values of these indices are considered to be coordinates in Euclidean space we defined analogue years to be those years whose positions are within a distance of 0.2 units in the Euclidean space from the position of the forecast year."

Additionally, we have added a line directly after the sentence in question giving further information as discussed above. This line reads as follows:

"The threshold is a compromise between being small enough to ensure that the climate setup is indeed similar to the year in question and being large enough to actually be able to identify some analogues from the historical ensemble."

Line 29 and following: Is there a reference (needed) for the physical support of the asymmetric weighting?

**25)**

The sentences in the manuscript directly following this line (page 5, line 29 – page 6, line 2) give our account for this physical support. For example, we explain that the relative importance of the snowpack earlier in the season is less than it is later in the season with respect to the coming meteorological conditions.

**26)**

We have corrected the typo

**27)**

Yes, we have clarified this by changing the sentence to now read:

"This process is repeated n times to give a validation dataset of length n, for this work n=35."

Line 27: Why is the relative mean absolute error used (error divided by SFV0y)? Would the absolute error (without the division) not be a stronger focus on the flood peaks and therefore more beneficial for the assessment of the skill of forecasting the spring flood? Please explain.

**28)**

The division operation converts the error form a volume to a ratio of the observed volume. This does not alter the relative emphasis of the metric, but it does make it more intuitive.

Line 28: The superscripts suggest that it is SFV to the power of y. A different notation is clearer.

**29)**

Another notation would be clearer from a mathematical understanding, however this notation is fairly common in the hydrology literature. Additionally, by retaining the current notation we are maintaining continuity with the previous works which this work builds on.

Line 6: Subbasin.

**30)**

Please see our response (5$^{th}$) to reviewer 1.

Line 6: There should be a reference to figure 3 included in the sentence.

**31)**

We added a cross reference to figure 3 at the end of the sentence.

**32)**

Line 10: How is total runoff divided between the 3 subbasins? Would it be interesting to use the new setup also for the two other subbasins?

The three clusters S1, S2, and S3 are made up of subbasins from seven river systems. They are not subbasins in themselves. Table 2 gives some basic statistics regarding the SFV in the subbasins of the different clusters. The prototype is aimed primarily at reservoir operators in the hydropower industry and the majority of the large operations are based in these three clusters. We agree that it would be interesting to apply this approach to the other two clusters but this will have to wait for now.

**33)**

Please see our response (6[th]) to reviewer 1.

**34)**

Line 27: We don't understand why the subbasins with 61, 63, and 73 percent of missing data are also included in the 84 basins that are investigated. Has it been checked whether and how the results change when excluding those three subbasins?

The inclusion of these basins is due to them being part of the current operational forecast system (see page 10, line 29 – line 29) and will be required going forward. We checked how their inclusion affected the results. Their inclusion typically reduced the apparent added value of the prototype over the current operational forecast system due to the need to use this data to train the statistical model. However, the reductions in the skill scores were not statistically significant.

**35)**

Line 31: What is the PTHBV dataset from SMHI? is this the ptq file for HBV (but without the q)?

Yes. PTHBV is the name of a gridded product for Sweden of P and T observations which is used to populate the ptqw file in HBV. The Q and W values are populated using station data.

**36)**

Page eleven:

Line 4: Do you refer to the performance measures for the HBV rainfall-runoff model of the historical data?

Yes, these are validation scores for HBV using perfect hindcasts i.e. forced using observed P and T.

Line 7: The 'than' near the end of the line is missing one or two adjectives describing latitude and elevation. Decide on either and or or.

**37)**

We have reworded this line to now read:

"Subbasins in cluster S1 are typically at a latitude or elevation lower than those in clusters S2 and S3, similarly the subbasins in S2 with respect to those in S3."

Line 22-24: We don't really understand where the 15 and 51 come from. We probably missed that somewhere, or is it not explained anywhere?

**38)**

These are the number of ensemble members that are available in the seasonal forecasts/hindcasts from the ECMWF. We have reworded page 11, line 23 to make this clearer in the context of the surrounding paragraph. It now reads:

"This is because the number of ensemble members available in the ECMWF seasonal forecast is limited to 15 for the hindcast period while the operational seasonal forecast ensemble has 51 members."

Page thirteen:

Line 21: Are these 5% of the catchments located in a certain area or do they have similar characteristics? Large/small, steep/flat, northern/southern?

**39)**

As we are more interested in the overall performance we did not put an emphasis on decomposing the possible reasons behind why the results for specific stations may have performed the way they did. Your questions are valid and should be investigated going forward, but in the context of this work it is less important. However, results from both this work and previous work by Olsson et al. (2016) suggest that the subbasins where the prototype does not perform as well tend to be located in, but not isolated to, the middle and lower reaches of the rivers. Also, please see our earlier response (no. 7).

Page fourteen:

Line 29: Why was exactly this site choose for the analysis of the forecast ensemble sharpness? Just one sentence, why exactly this basin is relevant out of the whole set of the 84 sites.

**40)**

The following sentence was included after page 14, line 29. It reads:

"This basin was chosen as an example of a where the prototype showed typical performance results i.e. neither the best nor the worst."

**41)**

Page sixteen:

Line 5-18: This is more discussion than real conclusion. Move?

We moved the parts that were more discussion in nature to the results and discussion section. This section now reads:

"In this paper we present the development and evaluation of a hydrological seasonal forecast system prototype for predicting the SFV in Swedish rivers. Initially, two versions of the prototype, MEads and MEhds, were evaluated together with the HE using climatology as a reference to both help select which version of the prototype to proceed with and to get a general impression of their skill to forecast the SFV. Thereafter the chosen prototype was evaluated using HE as a reference and finally the sharpness of the hindcast ensembles were analysed.

The main findings are summarized below:

• The prototype is able to outperform the HE approach 57% of the time on average. It is at worst comparable to the HE in forecast skill and at best clearly more skilful.

• The prototype is able to reduce the forecast error by 6% on average. This translates to an average volume of 9.5 x 106 m3.

• The prototype is generally more sensitive to uncertainty, that is to say that the ensemble spread tends to be more correlated with the forecast error. This is potentially useful to users as the ensemble spread could be used as a measure of the forecast quality.

• The prototype is able to improve the prediction of above and below normal events early in the season."

**42)**

Figures:

Figure 2: In the figure description it is mentioned that the spring flood is in the period between the onset and the last day of July. On the x-axis the days since the 1st January are written. We think that for the reader it would be easier if on the x-axis the first of the months were labelled and the date of the 80 days threshold was indicated (20th or 21st of March).

We disagree with the need to change the x-axes labels to dates. We feel that by doing so would complicate the figure more than it simplifies it. Figure 2 is meant as a generalised schematic showing the concept of how we define the spring flood period and not meant to give specific dates to the readers which are not mentioned elsewhere in the manuscript. The only deviation from what is mentioned in the body of the manuscript is that the 31$^{st}$ July is not explicitly.

As a compromise we have added the day numbers for the 31$^{st}$ July in parentheses the date is mentioned in the figure description. It now reads:

"The spring flood is the period between the onset and the last day of July (day 211/212 since the 1st January)."